# Effects of Safety State Augmentation on Safe Exploration

**Aivar Sootla**
Byju's Lab
aivar.sootla@gmail.com

**Alexander I. Cowen-Rivers**
Technische Universität Darmstadt
mc_rivers@icloud.com

**Jun Wang**
University College London
jun.wang@cs.ucl.ac.uk

**Haitham Bou Ammar**
Huawei R&D
haitham.ammar@huawei.com

## Abstract

Safe exploration is a challenging and important problem in model-free reinforcement learning (RL). Often the safety cost is sparse and unknown, which unavoidably leads to constraint violations — a phenomenon ideally to be avoided in safety-critical applications. We tackle this problem by augmenting the state-space with a safety state, which is nonnegative if and only if the constraint is satisfied. The value of this state also serves as a distance toward constraint violation, while its initial value indicates the available safety budget. This idea allows us to derive policies for scheduling the safety budget during training. We call our approach Simmer (Safe policy IMproveMEnt for RL) to reflect the careful nature of these schedules. We apply this idea to two safe RL problems: RL with constraints imposed on an average cost, and RL with constraints imposed on a cost with probability one. Our experiments suggest that "simmering" a safe algorithm can improve safety during training for both settings. We further show that Simmer can stabilize training and improve the performance of safe RL with average constraints.

## 1   Introduction

Reinforcement learning (RL) is a framework for sequential decision-making that makes minimal prior assumptions about the environment where the agent has to act or make the decisions [45]. The policy for taking actions is learned through interactions with the environment over time. RL has seen recent successes in playing video games with a computer [32], board games with a human [41] and is on a path toward real-life applications such as video compression [46], and plasma control [20]. There are still, however, some unsolved challenges for a successful deployment of RL such as efficient learning of constrained or safe Markov Decision Processes (MDPs) [4]. The constraints are typically modeled by a discounted sum of nonnegative costs that have to be smaller than some pre-defined value we call *the safety budget*.

Exploration is a crucial component of RL and is still an active area of research [25, 40, 34]. In the context of safe RL, while exploring, we do not want to incur the safety cost and constraint violations, making exploration a harder task. We can distinguish two main research directions for minimizing constraint violation in safe RL: a model-based approach (which includes partially and fully known environments) and a model-free approach. In the model-free case, which we focus on, the agent would almost certainly visit unsafe regions to learn the safe policy, and therefore it is next to impossible to avoid constraint violations completely. The general goal, in this case, is to minimize the number of violations during training. For example, [10] use a conservative safety critic and rejection sampling to choose

36th Conference on Neural Information Processing Systems (NeurIPS 2022).

a "safer" action, [48] design a curriculum learning approach, where the teacher resets the student violating safety constraints, and [21] learns to reset the policy if the safety constraint is violated [21].

In this work, we aim to improve model-free safe reinforcement learning by augmenting the state-space with one state encapsulating the safety information. This safety state is initialized with the safety budget and the value of the safety state can serve as a measure of distance to the unsafe region. This form of the safety state was proposed in [18] and introduced in RL in [43], where a safe RL algorithm with probability one constraints was derived. In this paper, we take a closer look at the purposes of the safety state in a broader context. First, we claim that safety state augmentation *is often crucial for the averaged constrained problem* as well and provide examples of such occurrences. In some particular problems, the use of safety state augmentation may potentially be avoided, however, this can be said about any state in the environment. Second, our main suggestion is that scheduling the safety budget at different training epochs can improve the algorithm's performance. Our claim extends equally over averaged and almost surely constrained reinforcement learning problems. In particular, we achieve state-of-the-art performance in some environments in the safety gym benchmark. Further, we claim that scheduling the initial safety budget can lead to reducing safety constraint violations during training for safe RL with probability one constraints. We design two algorithms that automatically tune the safety budget, one is based on a classical control engineering PI controller, while the other uses Q learning to decide on the safety budget.

*Related work.* Many of the current safe RL methods are extensions of the most successful RL algorithms: Trust region policy optimization (TRPO) [38], proximal policy optimization (PPO) [39], soft actor-critic (SAC) [23] etc. Safe versions of TRPO, PPO, and SAC with a Lagrangian approach were first presented in [37], which is still considered to be one of the major baselines. A direct extension of TRPO by adding constraints to the trust region update was proposed in [2]. Model-based approaches were also considered cf. [36, 16, 27, 30] and most of them took a Bayesian approach to model one-step transitions. To our best knowledge, the most successful approaches for safe reinforcement learning to date are PID-Lagrangian [44] and LAMBDA [5]. The former views the Lagrangian multiplier update as another control problem and employs a PID controller to solve it (cf. [6]). Specifically, the authors link the multiplier update to integral control and add proportional and derivative controllers to achieve a superior behavior. On the other hand, [5] is a model-based approach that uses Bayesian world models to enhance safety. A recent work [29] formulated safe RL as inference resulting in a sample efficient off-policy approach.

Other formulations of safe RL were considered in the literature. For example, [14, 16, 51] proposed to use conditional-value-at risk (CVaR) constraints, while [43, 12] proposed to enforce constraints with probability one. Further, as we discussed above eliminating the number of constraint violations typically requires strong assumptions, e.g., finite state space [47], [50] [42], knowledge of a partial model [28] or initial safe policy [8]. These results are in the spirit of safe RL with control-theoretic notions [15, 9, 33, 13, 3, 19, 22], which make significant prior assumptions to guarantee safety. Finally, the closest algorithm in the literature to our method is the curriculum learning approach to Safe RL from [48]. Due to space limitations we discuss this approach in detail in Appendix. We finally mention that [31] proposed a two-player framework with the cooperating task agent and safety agents, [17] proposed a safety layer that would be applied after the action is computed using a classical policy, [24] defined a probabilistic shield for safety. We also note [52, 26] that considered safe deployment of RL policies in real-life settings.

## 2 Simmer: Safe policy improvement for reinforcement learning

### 2.1 MDP with safety state augmentation

We consider a constrained reinforcement learning setting defined for a Markov Decision Process (MDP) $\mathcal{M} = \langle \mathcal{S}, \mathcal{A}, \mathcal{P}, r, \gamma_r \rangle$ with transition probability $\mathcal{P} : \mathcal{S} \times \mathcal{A} \times \mathcal{S} \rightarrow [0, 1]$ acting on state $\mathcal{S}$ and $\mathcal{A}$ spaces, with the reward $r : \boldsymbol{S} \times \boldsymbol{A} \times \boldsymbol{S} \rightarrow \mathbb{R}$ and the discount factor $\gamma_r \in (0, 1]$. The MDP is endowed with the following optimization problem

$$\max_{\pi(\cdot|\boldsymbol{s})} \mathbb{E} \sum_{t=0}^{T-1} \gamma_r^t r(\boldsymbol{s}_t, \boldsymbol{a}_t, \boldsymbol{s}_{t+1}), \text{ subject to: } g\left(d - \sum_{t=0}^{T-1} \gamma_l^t l(\boldsymbol{s}_t, \boldsymbol{a}_t, \boldsymbol{s}_{t+1})\right) \geq 0, \quad (1)$$

with the time horizon $T > 0$, the safety discount factor $\gamma_l \in (0, 1]$, the safety cost $l : \boldsymbol{S} \times \boldsymbol{A} \times \boldsymbol{S} \rightarrow [0, +\infty)$. The statistic $g(\cdot) : \mathbb{R} \rightarrow \mathbb{R}$ is a design choice (e.g., Mean, CVaR, chance

constraints etc [14]). In this paper, we consider two most relevant options, from our point of view: a) a constraint with probability one, i.e., $g_{\mathrm{po}}(z) = \mathbb{P}(z \geq 0) - 1$, b) a constraint on average $g_{\mathrm{av}}(z) = \mathbb{E}z$.

Similarly to [43], we augment the safety state information into the state-space by introducing the state $z_t = \gamma^{-t}(d - \sum_{k=0}^{t-1} \gamma_l^k l(\boldsymbol{s}_k, \boldsymbol{a}_k, \boldsymbol{s}_{k+1}))$, which has the following update $z_{t+1} = (z_t - l(\boldsymbol{s}_t, \boldsymbol{a}_t, \boldsymbol{s}_{t+1}))/\gamma_l$, with $z_0 = d$. Noting that the update is Markovian this state can be easily augmented into the MDP. The variable $z_t$ has the interpretation of the remaining safety budget, and by definition enforcing the constraint on $z_T$ is equivalent to enforcing the constraint on the accumulated cost. Now we can rewrite the safe RL problem as follows:

$$\max_{\pi(\cdot|\boldsymbol{s},z)} \mathbb{E} \sum_{t=0}^{T-1} \gamma_r^t r(\boldsymbol{s}_t, \boldsymbol{a}_t, \boldsymbol{s}_{t+1}), \text{ subject to: } g(z_T) \geq 0. \tag{2}$$

While [43] considered only the case $g_{\mathrm{po}}(z)$, we argue that the case of $g_{\mathrm{av}}(z)$ deserves additional attention in the context of safety state augmentation. For completeness, we review the main points of the approach [43] in Appendix. Note that the policy now also depends on the safety state $z$ and this feature deserves a more thorough discussion.

## 2.2  Do we need the safety state?

As a simple demonstration consider the cartoon in Figure 1. A robot needs to reach the goal while crossing the hazard region, which is marked by the red circle, and the safety cost is acquired for every time unit spent in the region. Both green and blue paths are safe, i.e., satisfy the constraint. However, at the crossing of these paths robot needs to know which path it took to this state. Switching from the green path to the blue one will lead to a constraint violation. Standard safe RL algorithms will have trouble with such scenarios while adding the safety state solves the issue. We confirm this observation in our experiments.

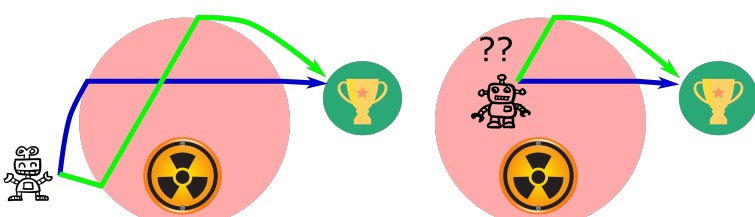

Figure 1: A robot needs to reach the goal while crossing the hazards region (marked by the red circle) and the safety cost is acquired for every time unit spent in the region. Both green and blue paths are safe, i.e., satisfy the constraint. However, at the path crossing robot needs to know which path it took to this state. Switching from the green path to the blue one will lead to constraint violations.

Let us now discuss how this logic can be mathematically formalized. In the deterministic case, this problem is well studied in the optimal control literature in the context of problems with known dynamics and with terminal or end-point constraints [49]. Further, the authors [43] showed that the policy dependence on the remaining safety budget is crucial for safe reinforcement learning with probability one constraints. The problem was also studied in the context of stochastic diffusions [35], where the authors derived the representation of the optimal value function. In both cases, it was shown that the optimal policy depends on the whole state including the safety state $z$. While in some cases one may not need to augment the safety state, in many situations it is critical.

## 2.3  Simmering Safe Reinforcement Learning

In this paper, we propose to use the initial safety budget $d$ as another tuning dial for the algorithms. Specifically, we will exploit the link between the safety budget $d$ with the initial state of the safety state. We argue that adjustment of $d$ during training from some initial value $d^{\mathrm{start}}$ to the target value $d^{\mathrm{target}}$ can lead to improved exploration in terms of safety and performance.

Let us now present the mathematical formulation. At every epoch $k$ we pick a test safety budget $d_k$, collect the data set $\mathcal{D}_k$, compute the returns $\mathbb{E}_{\mathcal{D}_k} \sum_{t=0}^{T-1} \gamma_r^t r(\boldsymbol{s}_t, \boldsymbol{a}_t)$ and the costs $\hat{g}_{\mathcal{D}_k}(z_T)$, where the function $\hat{g}_{\mathcal{D}_k}(\cdot)$ is the empirical mean or the maximum for the averaged and the probability one

constrained problems, respectively. Using this information we aim at solving the following problem at every epoch (but we only apply a pre-determined number of gradient steps):

$$\max_{\pi(\cdot|\boldsymbol{s},z)} \mathbb{E}_{\mathcal{D}_k} \sum_{t=0}^{T-1} \gamma_r^t r(\boldsymbol{s}_t, \boldsymbol{a}_t), \text{ subject to: } \hat{g}_{\mathcal{D}_k}(z_T) \geq 0, z_0 = d_k \tag{3}$$

where $s_0 \in \mathcal{S}_0$ and $z_0 = d_k$ are the initial states of the augmented MDP. Note that for off-policy algorithms the data set $\mathcal{D}_k$ can potentially grow with epochs, while for on-policy algorithms the data set $\mathcal{D}_k$ will be emptied on every epoch.

The key assumptions of our approach are as follows:

- There is a finite number of test safety budgets, i.e., we assume that $d_k$ can take the values $\{d_k^{\text{ref}}\}$ with $d_0^{\text{ref}} \leq \cdots \leq d_{K-1}^{\text{ref}}$ with $d^{\text{start}} = d_0^{\text{ref}}$ and $d^{\text{target}} = d_{K-1}^{\text{ref}}$;
- We assume that the values $\{d_k^{\text{ref}}\}$ are such that the task can be solved;
- At for every $k$ we perform only one epoch of optimization solving Problem 3.

The intuition behind our formulation is based on our empirical observations, where lower safety budgets usually caused lower bursts in accumulated safety costs. We hypothesize that the policy can quickly learn "extremely unsafe" actions thus providing low safety cost bursts for low safety budgets. Therefore if we start with a very strict safety budget $d^{\text{start}}$, then by gradually increasing the safety budget $d$ from $d^{\text{start}}$ to $d^{\text{target}}$, we can reduce the number of constraint violations during training. We will show that the problem of safety during training can be formalized as a two-level decision-making problem, however, even *a naïve approach of scheduling $d_k$ can lead to improvement in performance and safety*. We see this formulation as the first step toward eliminating safety violations during training.

## 2.4 Application: Simmering for safety during training

Exploration can often lead to constraint violation during training due to the inherent stochasticity of exploration. While there is a significant effort in research for safe exploration it typically requires significant prior assumptions on MDPs. We propose to embrace the philosophy of classical reinforcement learning and proceed with minimal assumptions. We propose to choose the test safety budget $d_k$ using another decision-making problem:

$$\max_{u_k \in [-\delta d, \delta d]} - \sum_k \text{ReLU}\left(-\hat{g}_{\mathcal{D}_k}(z_T^k(d^{\text{target}}))\right),$$
$$d_{k+1} = \text{clip}\left(d_k + u_k, d^{\text{start}}, d^{\text{target}}\right), d_0 = d^{\text{start}}, \tag{4}$$

where $z_T^k(d^{\text{target}})$ is the violation of the target constraint obtained as a result of solving one epoch of Problem 3, the action set is $[-\delta d, \delta d]$ and $\delta d$ is pre-determined, $\text{clip}(x, y, z)$ clips the value $x$ to a lower bound $y$ and an upper bound $z$, i.e., the function returns $y$ if $x \leq y$, $z$ if $z \leq x$ and $x$ otherwise.

We formalized our problem as reinforcement learning over a partially observable process with the non-stationary observations of the learning process. This two-level RL problem allows us to use a broad spectrum of tools available in sequential decision-making differing our approach from the existing in the literature. However, solving Problem 4 appears to be a daunting task, especially in the online setting as we intend. Hence, we will employ heuristic solutions, which may prove more effective than the quest to find an optimal solution. Intuitively, a gradual increase in $d_k$ by assigning scheduling increasing reference values (denoted by $d_k^{\text{ref}}$) would less likely lead to constraint violation due to stricter exploration constraints. However, fixing the schedule *a priori* limits the ability of the algorithm to react to constraint violations. To alleviate this issue we propose two approaches: PI Simmer (PI-controlled safety budget) and Q Simmer (Online Q-learning with non-stationary rewards). In both algorithms the intuition is again is based on our empirical observations that lower safety budgets cause lower bursts in accumulated safety costs. In particular, if the current accumulated costs are well below the safety budget $d_k^{\text{ref}}$, then we are very safe and the safety budget can be further increased. If the accumulated costs are around the safety budget, then we could stay at the same level or increase the safety budget. If the current accumulated costs are well above the safety budget $d_k^{\text{ref}}$, then the safety budget should be decreased.

**PI Simmer.** The idea for the PI controller is quite intuitive it takes the error term $e = d^{\text{ref}} - c$ and uses it for action computation. P stands for proportional control and links the error terms with actions

by multiplying the error term by the gain $K$. The proportional part delivers brute force control by having a large control magnitude for large errors, but it is not effective if the instantaneous error values become small. Proportional control cannot achieve zero error $e$ tracking. This happens since zero error results in a zero proportional action and hence the control over the error is lost. To deliver zero error, integral control is typically used, which sums up previous error terms and uses this sum to determine actions instead of the error. The integral term can be seen as an "action acceleration" (or "momentum") term which is large if the past errors are large. This can potentially cause unwanted behaviors (such as oscillations) if the corresponding gain $K_i$ is too large. Both gains $K$, and $K_i$ are usually chosen by tuning, but there are rules of thumb for tuning and choosing the gains, which we discuss in Appendix. PI controller can solve many hard control problems, but there are some implementation and engineering tricks and improvements. We introduce a simplified version in Algorithm 1 and provide a full version of the algorithm and our ablation studies in Appendix.

---

**Algorithm 1:** PI SIMMER (basic version)

---

**Inputs:** $\{d_k^{\mathrm{ref}}\}_{k=0}^{N-1}$ - safety budget schedule, hyper-parameters - $K_p$, $K_i$, $\delta d$.
Set $d_0 = d_0^{\mathrm{ref}}$;
**for** $k = 0, \ldots, N-1$ **do**

    Perform one epoch of learning for the safe policy with $d_k$ and get the statistic $\hat{g}\left(z_T(d_k)\right)$;
    Compute the error term $e_k = d_k^{\mathrm{ref}} - \hat{g}\left(z_T(d_k)\right)$;
    Compute action $u_k^{\mathrm{raw}} = \underbrace{K_p e_k}_{\text{P-Part}} + \underbrace{K_i \sum_{i=k-T_i}^{k} e_i}_{\text{I-Part}}$;
    Set $d_{k+1} = \mathrm{clip}(u_k^{\mathrm{raw}}, -\delta d, \delta d) + d_k$;

**end**

---

**Q Simmer.** Consider an MDP with the states $\{d_0^{\mathrm{ref}}, \ldots, d_{K-1}^{\mathrm{ref}}\}$, which for simplicity of notation we denote $\{0, \ldots, K-1\}$, with the actions $\{a_{-1}, a_0, a_{+1}\}$, where the action $a_{+1}$ moves the state $s = i$ to the state $s = i + 1$, $a_{-1}$ moves the state $s = i$ to the state $s = i - 1$ and the action $a_0$ does not transfer the state. Note that the action $a_{+1}$ is defined for all $i < K - 1$ and the action $a_{-1}$ is defined for all $i > 0$. Our design for the rewards of this MDP is guided by the following intuition.

$$
\begin{array}{ccc}
\text{We are not safe} & \text{We are borderline safe} & \text{We are very safe} \\
\text{if } s - o \leq -\delta: & \text{if } |s - o| \leq \delta: & \text{if } s - o \geq \delta: \\[4pt]
r = \begin{cases} 2 & a = -1, \\ -1 & a = 0, \\ -1 & a = 1, \end{cases} &
r = \begin{cases} -1 & a = -1, \\ 1 & a = 0, \\ 1 & a = 1, \end{cases} &
r = \begin{cases} -1 & a = -1, \\ 1 & a = 0, \\ 2 & a = 1. \end{cases}
\end{array}
\tag{5}
$$

where $o = \hat{g}\left(z_T(d_k)\right)$ is a safety violation statistic (over an epoch) of the safe RL algorithm, $\delta$ is a significance violation threshold. We use a Q-learning update to learn the Q function:

$$
Q(s_t, a_t) = (1 - l_r)Q(s_t, a_t) + l_r(r_t + \max_b Q(s_{t+1}, b))
\tag{6}
$$

where $l_r$ is the learning rate and get the action with $\varepsilon$-greedy exploration strategy:

$$
a_t = \begin{cases} \mathrm{argmax}_b\, Q(s_t, b) & \text{with probability } \varepsilon, \\ \text{random} & \text{with probability } 1 - \varepsilon. \end{cases}
\tag{7}
$$

We summarize the approach as Algorithm 2.

# 3 Experiments

## 3.1 Baselines, environments and code base

**Environments:** We use the safe pendulum environment defined in [16], and we also use the custom-made safety gym environment with deterministic constraints, which we call static point goal [51]. In this environment with 46 states and 2 actions, a large hazard circle is placed before the goal and forces the agent to go around it to reach the goal similarly to our cartoon in Figure 1.

**Algorithm 2:** Q-SIMMER
___
**Input:** $\{d_k^{\mathrm{ref}}\}_{k=0}^{K-1}$ - an *a priori* chosen state space; hyper-parameters - $N, \tau, l_r, \delta, \varepsilon$;
Initialize $d_0 = d_0^{\mathrm{ref}}$
**for** $k = 0, \ldots, N - 1$ **do**

  Perform one epoch of learning for the safe policy with $d_k$ and get the statistic $\hat{g}\left(z_T(d_k)\right)$;
  Update the $Q$ function as in Equation 6 while computing the reward $r$ as in Equation 5;
  Compute an action $a_k$ as in Equation 7 and compute $d_{k+1}$ accordingly;

**end**
___

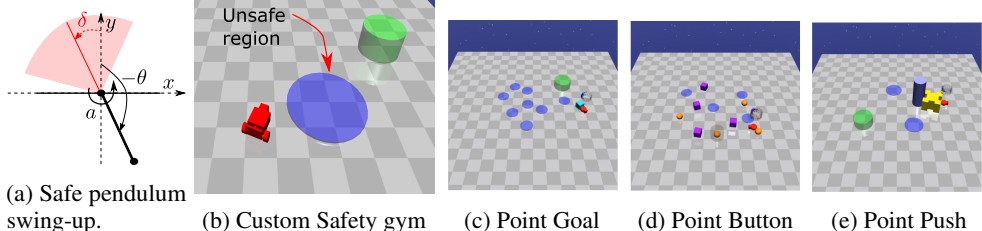

(a) Safe pendulum swing-up.  (b) Custom Safety gym  (c) Point Goal  (d) Point Button  (e) Point Push

Figure 2: (a): the safe pendulum environment ([16]). $\theta$ - is the angle from the upright position, $a$ is the action, and the angle $\delta$ defines the unsafe region position where the safety cost is the angle difference to $\delta$ and is incurred only in the red area. (b): the custom safety gym environment ([51]): robot needs to reach the goal while avoiding the unsafe region. (c)-(e): Safety Gym Goal, Button and Push tasks for the robot Point. Car robot environments are depicted in Appendix. The illustrations are from [43, 37].

We provide additional details in Figure 2 and Appendix. The rest of our tests are performed on the safety gym benchmarks [37].

**Code base:** Our code is based on two repositories: safety starter agents [37], and PID Lagrangian [44]. The code for PI Simmer and Q Simmer is available at `https://github.com/huawei-noah/HEBO/tree/master/SIMMER`. We use default parameters for both code bases unless stated otherwise.

**Computational resources:** We performed all computations on a PC equipped with 512GB of RAM, two Intel Xeon E5 CPUs, and four 16GB NVIDIA Tesla V100 GPUs.

**Baselines (●) and our algorithms (■):**
- CPO, Lagrangian PPO (L-PPO), and TRPO (L-TRPO); Standard baselines from [37];
- PID-Lagrangian. An algorithm stabilizing L-PPO learning [44];
- LAMBDA. A model-based method showing great performance on Safety Gym [5];
- PO-PPO PPO-based algorithm with probability one constraints from [43];
■ PI-Simmer - Scheduling safety budget using PI controller for PO-PPO;
■ Q-Simmer - Scheduling safety budget using Q learning for PO-PPO;
■ L-PPO (PID-L) w SA - L-PPO (PID-L) solving Problem 2 with safety state augmentation;
■ Simmer L-PPO (PID-L) - L-PPO (PID-L) w SA and safety budget scheduling.

### 3.2 Improving Safety During Training for Pendulum Swing-Up

Improving safety during training is more suited for almost surely safe RL and we will take PO PPO as our baseline. In this setting, we can aim to reduce the number of individual trajectories that violate the constraints, and thus we can avoid estimating the statistic $g$. For PI Simmer we chose the following hyper-parameters $K = 0.01$, $K_i = 0.005$, $K_{\mathrm{aw}} = 0.01$ and $\tau = 0.995$. The parameter $K_{\mathrm{aw}}$ resets the integral term avoiding the accumulation of error - the higher the value the more aggressive is reset. The parameter $\tau$ is the Polyak update for the error term, which pre-processes the error term for the PI controller. We discuss these parameters in detail in Appendix. We also note that except for an initial burst of violations both our approaches manage to keep the number of violations quite low. Overall we found that low values for $K$ and $K_i$ are beneficial to avoid overreaction to constraint violations. In this case, keeping the value of $K_{\mathrm{aw}}$ low is advisable as the action saturation does not occur too often. Finally, keeping $\tau$ close to one will force the controller to react to most of the constraint violations. For Q Simmer we chose $\delta = 1$, $\tau = 0.995$, $l_r = 0.05$, and $\varepsilon = 0.95$. It appears that a fast learning rate here

can allow for learning, sufficiently fast forgetting of past rewards, but also to avoid catastrophic forgetting. As we consider a finite state MDP we can avoid using sophisticated techniques for online learning and use the simplest one — tuning learning rate. We chose $\delta$ and $\tau$ to avoid frequent state transitions.

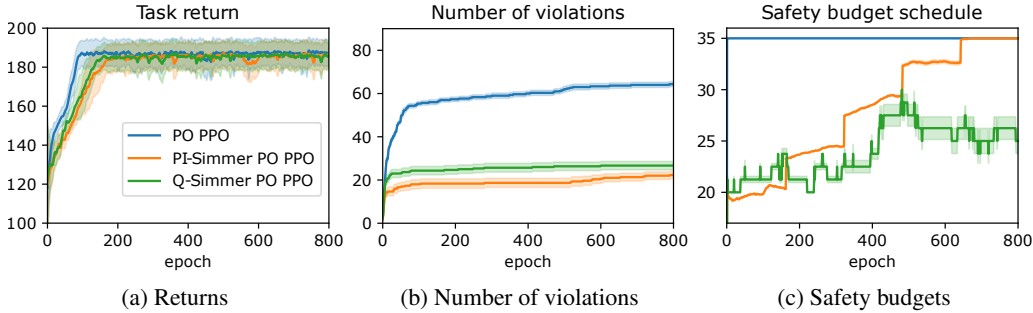

(a) Returns          (b) Number of violations          (c) Safety budgets

Figure 3: Reducing safety violations during training for Safe RL with constraints almost surely.

We compare our algorithm to PO PPO in terms of the number of trajectories with constraint violations, and returns, and compare the progression of the schedule $d_k^{\text{test}}$. Naturally, some individual trajectories still violate the constraint, but the number can be significantly reduced using Simmer RL as Figure 3 suggests. While PI Simmer outperforms Q Simmer in these runs, it is worth mentioning that PI Simmer uses more prior information than Q Simmer. Indeed, while composing a schedule is not hard, we still have to identify the switch points, which are learned by Q Simmer. We perform ablation studies on the parameters and discuss their choice in more detail in Appendix.

## 3.3 Guiding Exploration by Scheduling Safety Constraints

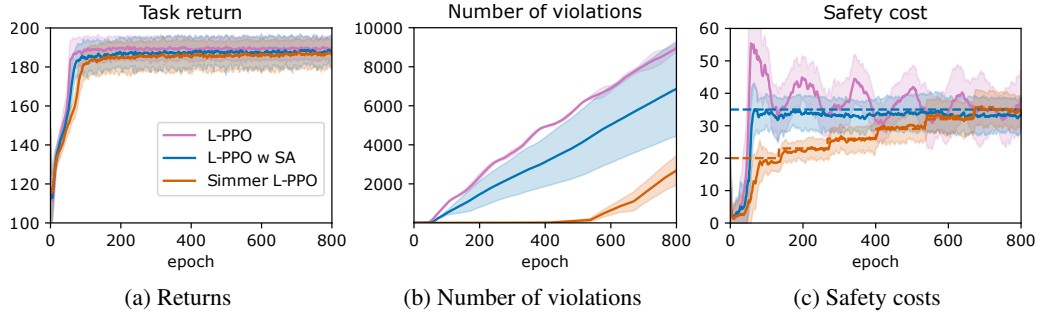

(a) Returns          (b) Number of violations          (c) Safety costs

Figure 4: Comparison of Simmer L-PPO, L-PPO with and without safety state augmentation. Mean returns and cost are computed over a hundred different trajectories obtained for three different seeds. Shaded areas represent standard deviations.

We now turn our attention to safe RL with constraints imposed on average costs. We test the performance of Simmer L-PPO and L-PPO w SA on the swing-up pendulum environment and present training results in Figure 4. Here we use safety starter agents as a base learner for all algorithms. The major observations are that L-PPO w SA delivers *almost no constraint violations* with respect to the mean cost estimate, and therefore using Q Simmer and PI Simmer is not necessary. In the meantime, L-PPO has even trouble converging. Note that we have used *the same hyper-parameters for all algorithms*, which are default parameters in safety starter agents and the learning rate $0.03$. While the behavior of the L-PPO algorithm can certainly be improved with tuning, we note that simply augmenting the safety state leads to improved performance as well as stability of the algorithm. Further, we observe that Simmer L-PPO leads to a fewer number of violations, however, the rate of violations for the safety budget of $35$ is fairly similar to L-PPO w SA.

Interestingly, a similar picture occurs with more advanced baselines such as PID Lagrangian [44] and more complicated environments. Here, we used the static point goal environment designed in [51], where a large static hazard region is placed in front of the goal, which is similar to our motivational example in Figure 1. The results for PID-L and PID-L with state augmentation are depicted in Figures 5a and 5b, respectively, suggest that the presence of the safety state stabilizes training and leads

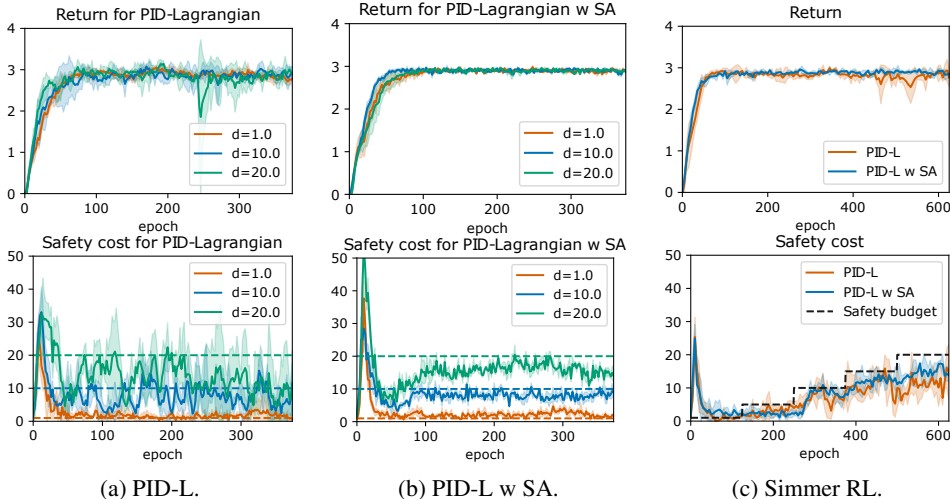



(a) PID-L.        (b) PID-L w SA.        (c) Simmer RL.

Figure 5: PID-L with and without safety state augmentation. $d$ is the safety budget used in training. The curves are means and shaded areas are standard deviations computed over 3 runs. These results suggest that safety state augmentation can stabilize training and deliver safer solutions.

to a more consistent constraint satisfaction for different safety budgets $d$. Note that hyper-parameters for all the runs are the same for both algorithms. We further apply the naïve simmer approach to both baselines with results depicted in Figure 5c. In both cases, the safety budget takes values of 1, 5, 10, 15, and 20, and increased after equal time intervals. Note that in both cases now training curves are quite stable, although state augmentation delivers an extra boost. In all our experiments we used the same hyper-parameters for all versions of PID-L, i.e., $K = 0.1$, $K_i = 0.01$, $\gamma_l = 0.99$.

Overall, our experiments suggest that safety during training with constraints imposed on average costs becomes a much easier problem with safety state augmentation. Indeed, even intuitively every outlier trajectory with a large constraint violation should bias the average cost and should instruct the algorithm not to follow this path. We note that in both cases above we have sparse costs, i.e., the agent encounters unsafe regions and incurs costs while navigating toward the goal. While an algorithm without the safety state will receive information on constraint violation after an episode, the safety state would constantly inform the algorithm of the distance toward a violation. This is one of the reasons why safety state augmentation can learn the task better. Note that simmering additionally offers fewer constraint violations while training. These results suggest that simmering safe RL together with state augmentation delivers overall safer solutions with stable training curves.

### 3.4 Tests on safety gym benchmarks

We finally test our approach on more challenging environments from the safety gym benchmark: PointPush1, PointGoal1, PointButton1 in Figures 6a, 6b, and 6c, respectively, and CarPush1, CarGoal1 and CarButton1 in Figures 7a, 7b, and 7c, respectively. The hyper-parameters and the tuning specifics can be found in Appendix. Again we use a naïve schedule increasing the safety budget from 10 to 25 in 5 unit increments. We observe that Simmer PID-L and PID-L outperform other baselines significantly in terms of the return while delivering safe policies. Simmer PID-L does perform very similarly to PID-L in terms of returns but generally outperforms PID-L in terms of safety costs and cost rates. We note that our results for PointGoal1 are consistent with the results from [44].

## 4 Conclusion

We augment the safety information into the state space and show how we can effectively use it to improve safe exploration. The safety state is nonnegative if and only if the constraint is satisfied and therefore it can serve as a distance toward constraint violations. We argue that the optimal policy must depend on this information to achieve safe performance. We validate this argument using

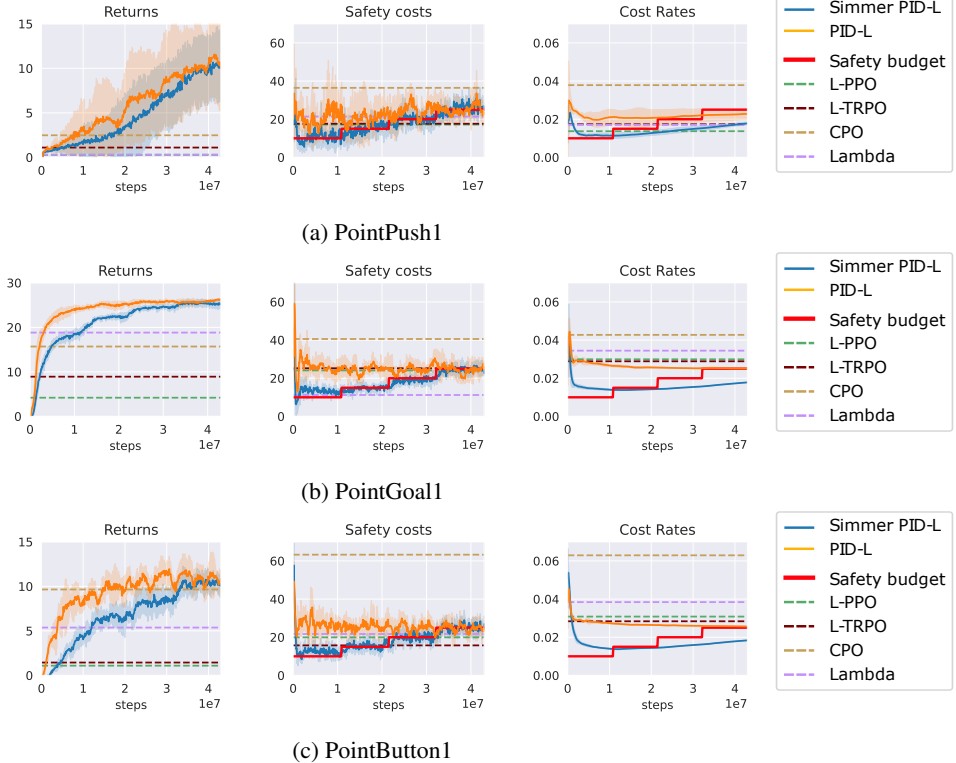

(a) PointPush1

(b) PointGoal1

(c) PointButton1

Figure 6: Results for the point robot in the goal, button, and push environments. Simmer PID-L outperforms all baselines in terms of the cost rate, which is a measure of safety during training. At the same time, Simmer PID-L is competitive with PID-L in terms of the returns or safety costs, while significantly outperforming other baselines. The curves are means and shaded areas are variances computed over 5 runs with random seeds.

intuitive examples, references to theoretical results, and experiments. We do not doubt that safety state augmentation is needed for effective safe RL.

Safety state augmentation and simmering show superior performance on pendulum swing-up and static point goal tasks for average constrained problems. We further tuned PID Lagrangian to show very strong performance on safety gym benchmarks, which has not been previously reported. Simmer PID Lagrangian shows competitive performance in terms of returns to PID Lagrangian and outperforms all baselines (including PID Lagrangian) in terms of the cost rate and the costs. Further, using state augmentation for the average constrained problems appears to be quite beneficial as well. Scheduling the safety budget can stabilize safe algorithms without state augmentation, but using both state augmentation and simmering improves performance and safety. We achieve this performance and safety boost at the expense of sample efficiency since we effectively learn a series of safe policies.

Simmering RL algorithms with probability one constraints can significantly reduce safety violations during training. We illustrate this feature by developing two algorithms for safe learning. The first one is based on a PI controller and allows for the adjustment of a pre-defined learning schedule depending on the estimated training cost. The second one is based on online Q learning and learns to adjust the safety budget automatically. Both approaches have some advantages and limitations. PI simmering is more effective if there is a reasonable safety budget schedule, while Q simmering requires less prior knowledge to learn. We have tested PI-Simmer and Q-Simmer on a rather simple in terms of the state and action spaces environments. We foresee that in a more complex setting the main problem would be learning the agent's behavior, but our algorithmic development would remain valid. Nevertheless, it would be interesting to extend this approach to a more complex setting.

Our algorithmic development is focused on probability one constraints for two reasons. First, since probability one constraints have to be imposed *on all trajectories* the empirical estimate of violations

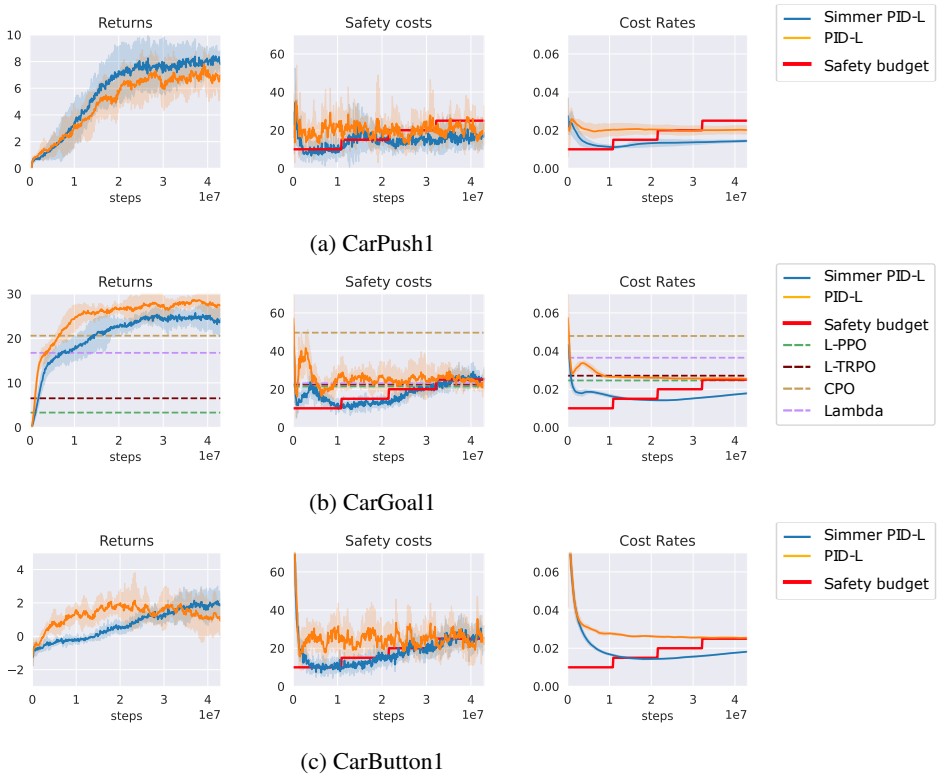

Figure 7: Results for the car robot in the goal, point and push environments. Simmer PID-L outperforms all baselines in terms of the cost rate, which is a measure of safety during training. At the same time, Simmer PID-L is competitive with PID-L in terms of the returns or safety costs, while significantly outperforming other baselines. The curves are means and shaded areas are variances computed over five runs with random seeds. Note that we have baseline results only for the CarGoal1 environment.

is simpler than in the average constrained case (i.e., maximum vs empirical average). Second, it appears that in the average constrained case with "simple" environments (such as pendulum swing-up and static point goal), we can track the constraint quite efficiently after the initial (and unavoidable) burst in constraint violations. Our experiments suggest that PI Simmer and Q Simmer can be redundant for the average constrained RL.

Augmenting the safety state and scheduling the safety budget does not solve all the problems in safe exploration. First, an initial burst of constraint violations is an inevitable reality of using our model-free approach. It appears that lowering the initial safety budget lowers the initial burst of constraint violations, but does not completely solve the problem. We do not see how to address this limitation without making further assumptions about the environment. Second, it is noticeable that the performance on the PointPush1 environment is quite noisy for both PID-L and Simmer PID-L. This is because some seeds achieve very good performance (return of 15) and some seeds do not (return of 5), while the learning curves appear to be stable. This suggests that the learning procedure finds local maxima. In our point of view, this calls for a more sophisticated algorithm for exploration maximizing return, which may help achieve stable performance in this environment. Beyond these limitations, one can also list safety violations caused by variance and disturbances, lack of provable safety guarantees, etc. Studying these limitations is outside of the scope of this paper.

# 5 Acknowledgement

This work was performed while the first two authors were with Huawei R&D UK. The authors thank Dr. Zimmer from Huawei R&D UK for helpful discussions and suggestions, as well as, the anonymous referees whose suggestions and comments significantly improved the paper.

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
