# Appendices

## A1 Related work

### A1.1 Safe Reinforcement Learning Using Curriculum Induction

Here we review the work from [48]. Consider the Safe RL problem $\mathcal{M} = \langle \mathcal{S}, \mathcal{A}, \mathcal{P}, r, \mathcal{D} \rangle$ with the following objective:

$$\max_{\pi} \quad \mathbb{E}_{\rho^{\pi}} \sum_{t=0}^{T} r(\boldsymbol{s}_t, \boldsymbol{a}_t, \boldsymbol{s}_{t+1})$$

$$\text{s.t.} \quad \mathbb{E}_{\rho^{\pi}} \sum_{t=0}^{T} \mathbb{I}(\boldsymbol{s}_t \in \mathcal{D}) \leq \kappa$$

where $\rho^{\pi}$ is the distribution of trajectories induced by $\pi$, $\mathbb{I}$ is the indicator function and $\mathcal{D}$ is the unsafe set. Note that this safe RL problem is less general than the standard formulation of safe RL.

The authors introduce a teacher-student hierarchy. The student tries to learn a safe policy, while the teacher is guiding the student through interventions $\mathcal{I}$. The interventions $\mathcal{I}$ are represented by pairs $\langle \mathcal{D}_i, \mathcal{T}_i \rangle$ that modify the safe RL problem into a student's problem $\mathcal{M}_i = \langle \mathcal{S}, \mathcal{A}, \mathcal{P}_i, r_i, \mathcal{D}, \mathcal{D}_i \rangle$, where we make the following modifications to the safe RL problem. The state transitions are modified as $\mathcal{P}_i(\boldsymbol{s}'|\boldsymbol{s}, \boldsymbol{a}) = \mathcal{P}(\boldsymbol{s}'|\boldsymbol{s}, \boldsymbol{a})$ for all $\boldsymbol{s} \notin \mathcal{D}_i$ and $\mathcal{P}_i(\boldsymbol{s}'|\boldsymbol{s}, \boldsymbol{a}) = \mathcal{T}_i(\boldsymbol{s}'|\boldsymbol{s})$ for all $\boldsymbol{s} \in \mathcal{D}_i$. This means that the teacher changes the probability transition for the student if they enter the set $\mathcal{D}_i$. The reward is modified as well: $r_i(\boldsymbol{s}, \boldsymbol{a}, \boldsymbol{s}') = r(\boldsymbol{s}, \boldsymbol{a}, \boldsymbol{s}')$ if $\boldsymbol{s} \notin \mathcal{D}_i$ and $r_i(\boldsymbol{s}, \boldsymbol{a}, \boldsymbol{s}') = 0$ if $\boldsymbol{s} \in \mathcal{D}_i$. Therefore, the student does not get any reward in the unsafe set. The student incorporates the interventions into their objective as follows:

$$\max_{\pi} \quad \mathbb{E}_{\rho_i^{\pi}} \sum_{t=0}^{T} r_i(\boldsymbol{s}_t, \boldsymbol{a}_t, \boldsymbol{s}_{t+1}),$$

$$\text{s.t.} \quad \mathbb{E}_{\rho_i^{\pi}} \sum_{t=0}^{T} \mathbb{I}(\boldsymbol{s}_t \in \mathcal{D}) \leq \kappa_i,$$

$$\mathbb{E}_{\rho_i^{\pi}} \sum_{t=0}^{T} \mathbb{I}(\boldsymbol{s}_t \in \mathcal{D}_i) \leq \tau_i,$$

where $\kappa_i$ and $\tau_i$ are intervention-specific tolerances set by the teacher.

To learn the teacher's policy the following constraints are followed:

- The unsafe set is contained in the intervention set $\mathcal{D} \subseteq \mathcal{D}_i$
- If $\kappa_i + \tau_i \leq \kappa$, then the set of feasible policies of the student is a subset of the set of feasible policies of the safe RL problem $\Pi_{\mathcal{M}_i} \in \Pi_{\mathcal{M}}$.

The teacher learns when to intervene and to switch between different interventions. The teacher is modeled by a POMDP $\langle \mathcal{S}^T, \mathcal{A}^T, \mathcal{P}^T, \mathcal{R}^T, \mathcal{O}^T \rangle$, where the state-space is the set of all student policies $\mathcal{S}^T = \overline{\Pi}_{\mathcal{M}}$ (not only feasible ones), the action space is the space of all interventions $\mathcal{A}^T = \mathcal{I}$, i.e., the teacher chooses the index $i$ for the student problem, the state transitions $\mathcal{P}^T : \overline{\Pi}_{\mathcal{M}} \times \mathcal{I} \times \overline{\Pi}_{\mathcal{M}} \to [0,1]$ is governed by the student's algorithms, the observation space $\mathcal{O}^T = \Phi$ is the space of evaluation statistics of student policies $\pi$. Finally the reward function is defined through the policy improvement $\mathcal{R}^T(n) = \hat{V}(\pi_{n,i}) - \hat{V}(\pi_{n-1,i})$, where the index $i$ denotes the student. The total reward for an episode is $\hat{V}(\pi_{N_s,i}) = \sum\limits_{n=1}^{N_s} \mathcal{R}^T(n)$. Now the policy is transferred from the trial $n-1$ to the next $n$. Note also the student index $i$ can be understood as an episode of learning teacher's policy in this case.

The major difference of our work is its online nature, while [48] pre-train teacher's policies deciding the curriculum, we do not pre-train our safety budget schedules $d_k$ explicitly, but use rules-of-thumb to determine the parameters of our online learning procedures. Our approach is preferable when a new task needs to be learned with minimal prior information about it, while the method from [48] is preferable when the policy of choosing the constraints can be transferred from another task. We also note that we do not reset our environment, but let it train further if the constraints are violated. Hence our approach can further be improved by adding reset policies similarly to [48] and [21].

## A1.2 RL with probability one constraints

We have introduced the safety state to the environment as follows:

$$
\boldsymbol{s}_{t+1} \sim \mathcal{P}\left(\cdot \mid \boldsymbol{s}_t, \boldsymbol{a}_t\right)
$$
$$
z_{t+1} = (z_t - l(\boldsymbol{s}_t, \boldsymbol{a}_t))/\gamma_l
$$

This means we have

$$
\gamma_l z_{t+1} - z_t = -l(\boldsymbol{s}_t, \boldsymbol{a}_t)
$$

and

$$
\gamma_l^{t+1} z_{t+1} - \gamma^t z_t = -\gamma_l^t l(\boldsymbol{s}_t, \boldsymbol{a}_t),
$$
$$
\gamma_l^t z_t - \gamma^{t-1} z_{t-1} = -\gamma_l^{t-1} l(\boldsymbol{s}_{t-1}, \boldsymbol{a}_{t-1}),
$$
$$
\vdots
$$
$$
\gamma_l^1 z_1 - z_0 = -l(\boldsymbol{s}_0, \boldsymbol{a}_0),
$$
$$
z_0 = d.
$$

Now we can sum up the left and the right-hand sides and obtain:

$$
\gamma_l^{t+1} z_{t+1} = d - \sum_{k=0}^{t} \gamma_l^k l\left(\boldsymbol{s}_k, \boldsymbol{a}_k\right).
$$

This means that the constraint can now be rewritten as $g(z_T) \geq 0$ resulting in the following optimization problem:

$$
\max_{\pi(\cdot)} \mathbb{E} \sum_{t=0}^{T-1} \gamma_r^t r(\boldsymbol{s}_t, \boldsymbol{a}_t, \boldsymbol{s}_{t+1}), \tag{A1}
$$
$$
g(z_T) \geq 0.
$$

If we consider the problem with probability one constraints then as a constraint we get $z_T \geq 0$ almost surely (or equivalently with probability one). It was noted that $z_T \geq 0$ almost surely is equivalent to $z_t \geq 0$ for all $t \geq 0$. Now the reward is reshaped to get:

$$\widehat{r}(\boldsymbol{s}_t, \boldsymbol{s}_z, \boldsymbol{a}_t, \boldsymbol{s}_{t+1}, z_{t+1}) = \begin{cases} r(\boldsymbol{s}_t, \boldsymbol{a}_t, \boldsymbol{s}_{t+1}) & \text{if } z_t \geq 0, \\ -\Delta & \text{if } z_t < 0. \end{cases} \tag{A2}$$

This problem can be solved using off-the-shelf RL algorithms for any finite $\Delta$. Finally, it was noted in [43] that with $\Delta \to +\infty$, the problem converges to safe RL with probability one constraints.

The main difference to [43] is that we investigate the effect of the safety state on safety during training for both probability one constrained RL and average constrained RL.

## A2   Additional algorithm details

### A2.1   PI-Simmer

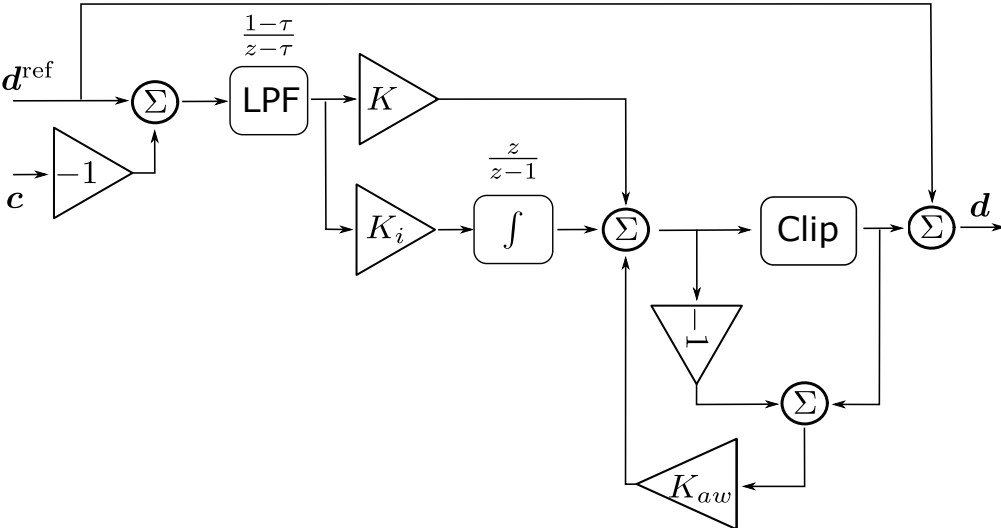

Figure A1: Block-diagram of our PI controller. The arrow signifies the direction of the signal, the triangles mean multiplication with a constant, and the rest of the block means the following operations: "LPF" stands for the low-pass filter (or the Polyak's update), "Clip" stands for clipping the values to a pre-defined minimum and maximum values, $\Sigma$ stands for a sum of signals and $\int$ stands for the integral of the signal over time. Main components of our controller: proportional gain $K$, integral gain $K_i$ for the integrator (marked as $\int$) anti-windup gain $K_{\text{aw}}$.

First, we discuss our design for the PI controller and discuss the necessary parts for it. Our main source for this discussion are the control engineering textbooks [7, 6]. The idea for the PI controller is quite intuitive: it takes the error term $d^{\text{test}} - c$ and uses it for action computation. P stands for the proportional control and multiplies the error term by the gain $K$ proportionally linking the error terms with actions. The proportional part delivers brute force control by having a large control magnitude for large errors, but it is not effective if the instantaneous error values become small. Proportional control cannot achieve zero error tracking, which is achieved by integral control and summing previous error terms. If the action values are clipped, however, integral control can lead to the unwanted phenomenon called *wind-up* and catastrophic effects [6]. This is specifically the case when the errors become large leading to large values of the integral, which in turn leads to saturated actions (actions take the clipped values) for a long time. There are several ways of dealing with wind-up, e.g., resetting the integral, and limiting the integration time, but we will take the feedback control approach where the previous saturation errors are fed back and used to determine the current action [6, 1]. Finally, we use a low pass filter for the error to avoid reacting to high-frequency fluctuations and acting only on the trends. The reader familiar with optimization algorithms may recognize the loss pass filter as the Polyak update.

---

**Algorithm A1:** PI SIMMER (Full version)

---

**Inputs:** $\{d_k^{\text{ref}}\}_{k=0}^{N-1}$ - safety budget schedule, hyper-parameters - $\tau$, $K_p$, $K_i$, $K_{\text{aw}}$, $\delta d$.

Set $d_0 = d_0^{\text{ref}}$;

**for** $k = 0, \ldots, N-1$ **do**

Perform one epoch of learning for the safe policy with $d_k$ and get the statistic $\hat{g}(z_T(d_k))$;

Compute and smooth the error term by $e_k = d_k^{\text{ref}} - \hat{g}(z_T(d_k))$, $\boldsymbol{w}_k = (1-\tau)\boldsymbol{w}_{k-1} + \tau e_k$;

Compute action: $u_k^{\text{raw}} = \underbrace{K_p \boldsymbol{w}_k}_{\text{P-Part}} + \underbrace{K_i \sum_{i=k-T_i}^{k} \boldsymbol{w}_i}_{\text{I-Part}} + \underbrace{K_{aw}(u_{k-1} - u_{k-1}^{\text{raw}})}_{\text{Anti-windup}}$;

Set $d_{k+1} = \text{clip}(u_k^{\text{raw}}, -\delta d, \delta d) + d_k$;

**end**

---

We propose the following update for the safety budget:

$$
\begin{array}{lll}
\text{Error term} & \text{Low-pass filter} & \text{P-part} \\
e_k = d_k^{\text{test}} - \hat{g}\left(z_T(d_k^{\text{test}})\right), & \boldsymbol{w}_k = (1-\tau)\boldsymbol{w}_{k-1} + \tau e_k, & P = K_p e_k, \\
\text{I-part} & \text{Anti-windup} & \text{Raw signal} \\
I = K_i \sum_{i=k-T_i}^{k} e_i, & AW = K_{aw}(u_{k-1} - u_{k-1}^{\text{raw}}), & u_k^{\text{raw}} = P + I + AW, \\
\text{Clipping} & \text{New safety budget} & \\
u_k = \text{clip}(u_k^{\text{raw}}, -\delta d, \delta d) & d_{k+1}^{\text{test}} = d_k^{\text{test}} + u_k, &
\end{array}
\tag{A3}
$$

where $e_k$ is the current constraint violation, $\boldsymbol{w}_k$ is the filtered error term $e_k$, $P$, $I$, $AW$ are the proportional, integral and anti-windup parts of the controller. The gains $K_p$, $K_i$, $K_{aw}$, as well as $T_i$, $\delta d$ and $\tau$ are hyper-parameters.

### A2.2 Q Simmer

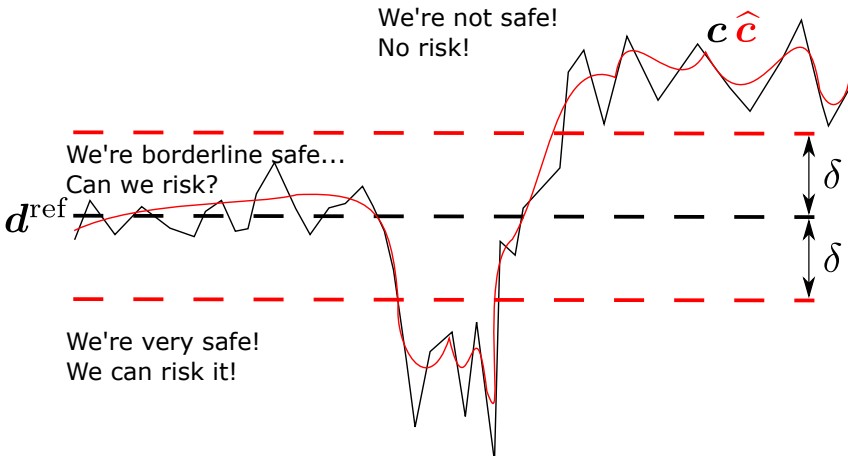

Figure A2: Intuition for Q simmer reward shaping: first the costs $\boldsymbol{c}$ are passed through a low-pass filter getting $\widehat{\boldsymbol{c}}$.

Our design for the Q learning approach is guided by the intuition depicted in Figure A2 and presented in what follows. If the current accumulated costs are well below the safety budget $d^{\text{ref}}$, then we are very safe and the safety budget can be further increased. If the accumulated costs are around the safety budget, then we could stay at the same level or increase the safety budget. If the current accumulated costs are well above the safety budget $d^{\text{ref}}$, then the safety budget should be decreased to ensure that the policy is incentivized to be safer.

Consider an MDP with the states $\{d_0, \ldots, d_{K-1}\}$, for simplicity of notation we consider the state space $\{0, \ldots, K-1\}$ with the actions $\{a_{-1}, a_0, a_{+1}\}$, where the action $a_{+1}$ moves the state $s = i$ to the state $s = i + 1$, $a_{-1}$ moves the state $s = i$ to the state $s = i - 1$ and the action $a_0$ does not transfer the state. Note that the action $a_{+1}$ is defined for all $i < K - 1$ and the action $a_{-1}$ is defined for all $i > 0$. The reward for this MDP is non-stationary and defined as follows:

$$
\begin{array}{llll}
\text{We are not safe} & \text{We are borderline safe} & \text{We are very safe} & \text{(A4)} \\
\text{if } s - o \leq -\delta : & \text{if } |s - o| \leq \delta : & \text{if } s - o \geq \delta : & \text{(A5)} \\
r = \begin{cases} 2 & a = -1, \\ -1 & a = 0, \\ -1 & a = 1, \end{cases} & r = \begin{cases} -1 & a = -1, \\ 1 & a = 0, \\ 1 & a = 1, \end{cases} & r = \begin{cases} -1 & a = -1, \\ 1 & a = 0, \\ 2 & a = 1. \end{cases} & \text{(A6)}
\end{array}
$$

where $o$ is the maximum accumulated cost (over an episode) of the constrained algorithm. We use a Q-learning update to learn the Q function:

$$Q(s_t, a_t) = (1 - l_r)Q(s_t, a_t) + l_r(r_t + \max_b Q(s_{t+1}, b)) \tag{A7}$$

and get the action with $\varepsilon$-greedy exploration strategy:

$$a_t = \begin{cases} \mathrm{argmax}_b Q(s_t, b) & \text{with probability } \varepsilon, \\ \text{random} & \text{with probability } 1 - \varepsilon. \end{cases} \tag{A8}$$

## A3  Implementation details

**Pendulum Swing-up.**  Our first environment is a safe pendulum swing-up defined in [16] and built upon the Open AI Gym environment [11]. The reward is defined as

$$r(s, a) = 1 - \frac{\theta^2 + 0.1\dot{\theta}^2 + 0.001a^2}{\pi^2 + 6.404},$$

while the safety cost

$$l = \begin{cases} 1 - \dfrac{|\theta - \delta|}{50} & \text{if } -25 \leq \theta \leq 75, \\ 0 & \text{otherwise,} \end{cases}$$

where $\theta$ is the deviation of the pole angle from the upright position. Effectively, we want to force the pendulum to swing up from one side. See Figure A3a.

**Safety Gym.**  We use several safety gym environments [37]. We use the benchmark environments PointGoal1, PointButton1, PointPush1, CarGoal1, CarPush1 (see Figures A3c-A3h), but we also use a custom-made one from [51] (the environment is schematically depicted in Figure A3b.). In the custom-made environment called static point goal, a large static hazard region is placed in front of the goal. This is similar to our motivational example in Figure 1. This environment confirms our intuition. We use two robots, the point robot has $46$ states and $2$ actions, while the car robot has $56$ states and $2$ actions.

**Code base and hyper-parameters.** Our code is based on two repositories: safety starter agents [37], and PID Lagrangian [44]. We have implemented the safety state augmentation following the description in [43]. We use default parameters for both code bases unless stated otherwise. We performed a parameter search where we set the batch size for all the environments to $512$. We then tried a few runs with $K_i \in [0.001, 0.005, 0.01, 0.05]$ and determined that $K_i = 0.05$ generally performs better for the goal and button tasks, while $K_i = 0.01$ appears to perform well for the push tasks. We then tested $K_p \in [0.01, 0.05, 0.1, 0.5, 1, 5]$ and our best results are presented in Table A1. The simmering schedule changes the safety budget every $200$ epochs, where the length of the epoch depends on the batch size and is presented in Table A1.

## A4  Further Experiments

### A4.1  Safety Gym

We present numerical results for the car and the point robot experiments in Table A2.

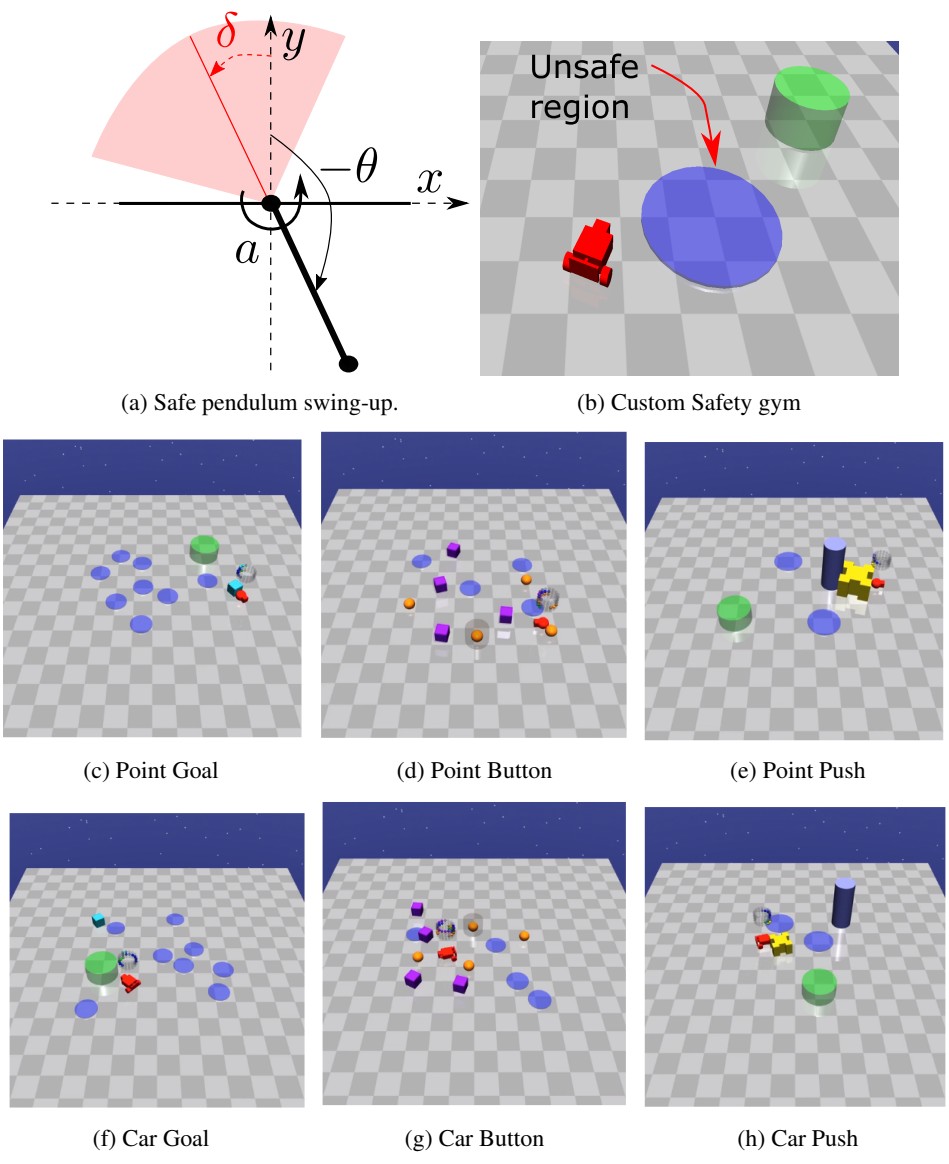

(a) Safe pendulum swing-up.

(b) Custom Safety gym

(c) Point Goal

(d) Point Button

(e) Point Push

(f) Car Goal

(g) Car Button

(h) Car Push

Figure A3: Panel a: safe pendulum environment from [16]. $\theta$ - is the angle from the upright position, $a$ is the action, and the angle $\delta$ defines the unsafe region position where the safety cost is the angle difference to $\delta$ and is incurred only in the red area. Panel b: a depiction of the custom safety gym environment from [51]: robot needs to reach the goal while avoiding the unsafe region. Panels c-h: Safety Gym Goal, Button, and Push tasks for robots Point and Car.

Table A1: Hyper-parameters for safety gym environments

| Parameter | Static PointGoal | PointGoal1 | PointButton1 | PointPush1 | CarGoal1 | CarButton1 | CarPush1 |
|---|---|---|---|---|---|---|---|
| Batch Size | 128 | 512 | 512 | 512 | 512 | 512 | 512 |
| $K_p$ (Vanilla) | 0.1 | 1 | 5 | 0.5 | 0.05 | 1 | 0.1 |
| $K_p$ (Simmer) | 0.1 | 5 | 5 | 0.5 | 0.1 | 1 | 1 |
| $K_i$ | 0.01 | 0.05 | 0.05 | 0.01 | 0.05 | 0.05 | 0.01 |
| $\gamma_c$ | 0.99 | 0.995 | 0.995 | 0.995 | 0.995 | 0.995 | 0.995 |
| Steps per epoch | 13312 | 53248 | 53248 | 53248 | 53248 | 53248 | 53248 |

Table A2: Numerical values of the results at the end of training. We compute the statistics over the last 15 epochs and present the mean $\pm$ standard deviation. We observe that the cost rates for Simmer are significantly smaller than the cost rates for PID Lagrangian

| | Simmer PID Lagrangian | | | PID - Lagrangian | | |
|---|---|---|---|---|---|---|
| Environment | Return | Cost | Cost rate ($\cdot 1e^2$) | Return | Cost | Cost rate ($\cdot 1e^2$) |
| PointGoal1-v0 | $25.24 \pm 0.81$ | $24.64 \pm 2.72$ | $\mathbf{1.77 \pm 0.01}$ | $26.23 \pm 0.22$ | $25.55 \pm 3.62$ | $2.51 \pm 0.00$ |
| PointButton1-v0 | $10.36 \pm 0.93$ | $24.37 \pm 2.81$ | $\mathbf{1.83 \pm 0.01}$ | $10.50 \pm 1.10$ | $23.98 \pm 3.73$ | $2.56 \pm 0.01$ |
| PointPush1-v0 | $10.27 \pm 3.45$ | $25.93 \pm 3.58$ | $\mathbf{1.77 \pm 0.06}$ | $10.50 \pm 4.41$ | $23.55 \pm 4.59$ | $2.28 \pm 0.26$ |
| CarGoal1-v0 | $23.97 \pm 2.43$ | $25.31 \pm 2.67$ | $\mathbf{1.78 \pm 0.02}$ | $\mathbf{27.46 \pm 2.35}$ | $22.68 \pm 3.77$ | $2.55 \pm 0.01$ |
| CarButton1-v0 | $\mathbf{1.93 \pm 0.64}$ | $25.95 \pm 5.39$ | $\mathbf{1.81 \pm 0.01}$ | $1.09 \pm 0.83$ | $24.98 \pm 6.53$ | $2.55 \pm 0.01$ |
| CarPush1-v0 | $\mathbf{8.01 \pm 0.64}$ | $17.60 \pm 5.48$ | $\mathbf{1.44 \pm 0.05}$ | $6.74 \pm 1.11$ | $19.41 \pm 6.10$ | $2.01 \pm 0.19$ |

## A4.2 Ablation for PI Simmer

First, we conduct the ablation study over the parameter of the low pass filter $\tau$ in Figure A4a, while we set $K = 0.1$, $K_i = 0.005$, $K_{\text{aw}} = 0.01$. We notice that smaller values of $\tau$ deliver small deviations from the schedule, while larger values have more freedom to decide on an appropriate schedule. However, it is noticeable that the runs with small $\tau$ (0.001 and 0.005) are constantly acquiring constraint violations, which happens less often for large values of $\tau$ (1 and 0.995). We believe this is because the controller with smaller values of $\tau$ ignores (filters out) spurious constraint violations, but with larger values of $\tau$ this is not happening. At the same time, PI Simmer with large $\tau$ changes the safety budget quite aggressively in the early stages due to constant constraint violations, which can be avoided if a small $\tau$ is chosen. Ultimately the choice of $\tau$ is up to the user, but we would recommend starting with large values.

We observed that it is probably most prudent to choose a small value for $K$ as our deviations from schedule $d_k^{\text{test}}$ are bounded and there is no need for drastic changes in the safety budget that proportional control can achieve. Indeed, we set $K_{\text{aw}} = 0.1$, $K_i = 0.005$, $\tau = 0.995$ and change the parameter $K$ and the results in Figure A4c supports our claims.

We now turn our attention to the gain $K_i$ of the integral control, while we set $K = 0.1$, $K_{\text{aw}} = 0.01$, $\tau = 1$. We stress that the choice of the gains $K_i$, $K_{\text{aw}}$ is somewhat interconnected, i.e., with increasing $K_i$ the gain $K_{\text{aw}}$ should increase as well to counteract saturation. However, it appears the choice of $K_i$ can have a more significant effect on performance. Figure A4b suggests that large values of $K_i$ lead to aggressive changes in the safety budget and, more importantly, later lead to saturation in control limiting the ability of PI Simmer to react to violations. Smaller values of $K_i$ allow for a balanced approach.

Finally, ablation over $K_{\text{aw}}$ in Figure A4d with $K = 0.1$, $K_i = 0.04$, $\tau = 1.0$ shows that while large values of $K_{\text{aw}}$ decrease the risk of saturation they also diminish the efficacy of integral control. On the other hand, with zero anti-windup gain, the actions are always saturated leading to more constraint violations without any reaction from the controller. Therefore, again a balanced approach is preferable.

## A4.3 Ablation for Q Simmer

We perform ablation on the following parameters the learning rate $l_r$, the Polayk's update $\tau$, and the reward threshold $r_{thr}$. Figure A5a suggest that large learning rates preferable. This is perhaps due to the ability to fast learning, but also fast forgetting may be important since the process is non-stationary. Ablation with respect to $\tau$ is performed in Figure A5b suggesting that higher values of $\tau$ are preferable, but $\tau = 1$ forces the algorithm to oscillate between two adjacent values too often.

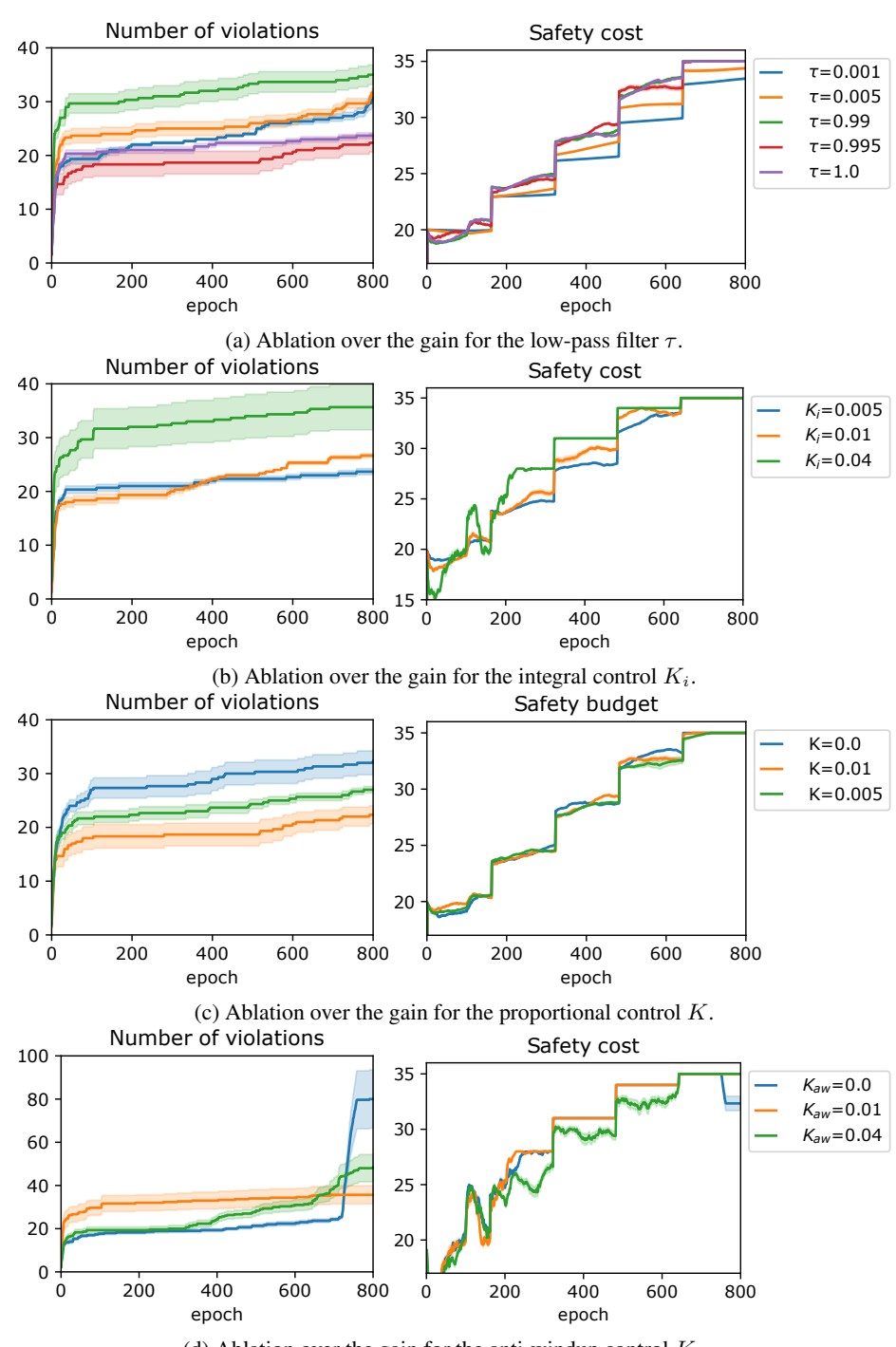

(a) Ablation over the gain for the low-pass filter $\tau$.

(b) Ablation over the gain for the integral control $K_i$.

(c) Ablation over the gain for the proportional control $K$.

(d) Ablation over the gain for the anti-windup control $K_{aw}$.

Figure A4: Ablation for PI Simmer

Similarly the value for $r_{thr}$ needs to be chosen so that oscillations between adjacent safety budget does not occur as Figure A5c suggests.

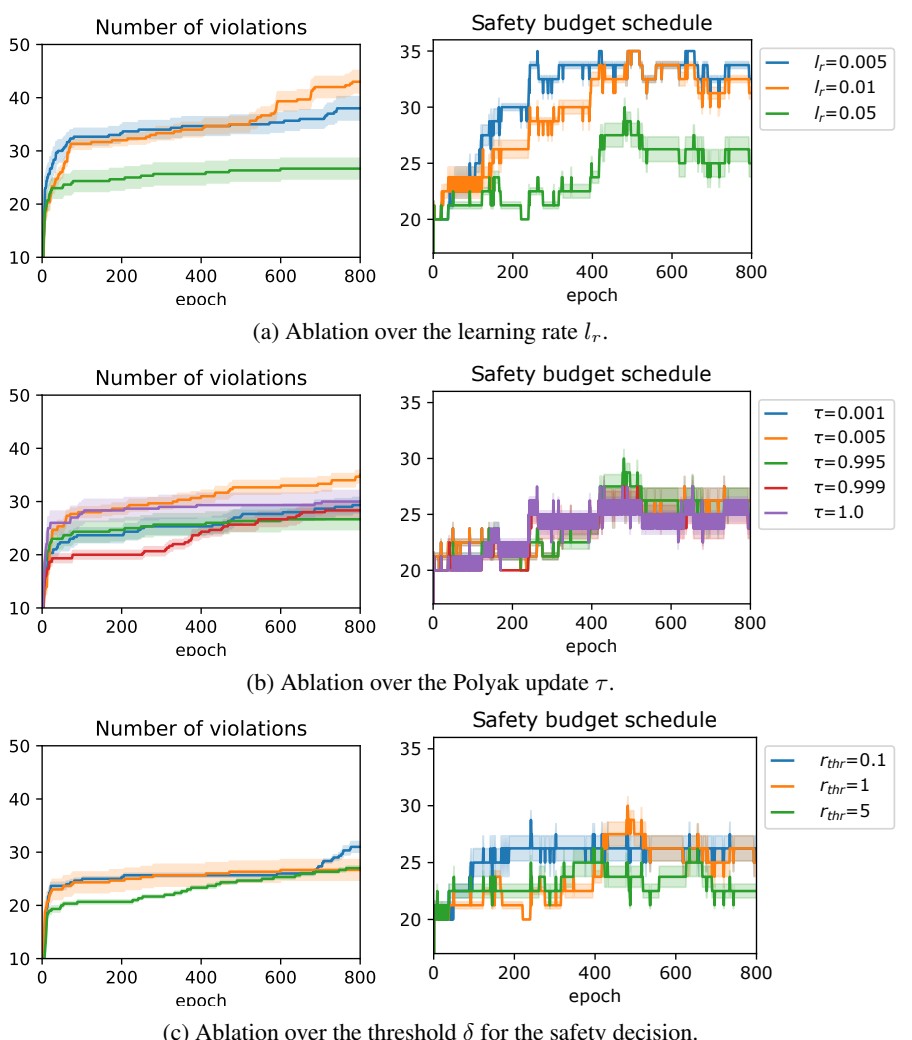

(a) Ablation over the learning rate $l_r$.

(b) Ablation over the Polyak update $\tau$.

(c) Ablation over the threshold $\delta$ for the safety decision.

Figure A5: Ablation for Q Simmer

## A5 Illustrations

Parts of the illustrations in Figure 1 are designed by gstudioimagen, macrovector official, Flaticon, and downloaded from `http://www.freepik.com`. The illustrations in Figures 2 and A3 are from [43, 37].