# OpenReview forum: "Enhancing Safe Exploration Using Safety State Augmentation"
_NeurIPS.cc/2022/Conference — NeurIPS 2022 Accept_

### Official Review · Reviewer_UM3D · 2022-06-30

**Rating:** 7
**Confidence:** 3
**Soundness:** 3 good
**Presentation:** 2 fair
**Contribution:** 3 good

**Summary:**

This paper explores learning a safety-state conditioned policy, along with methods for controlling the safety input adaptively to give extra slack to the policy for being more unsafe when it can afford to do so. I mainly have issues with parts of the writing of the paper, and some unanswered questions, that if fixed would make me recommend this paper for acceptance.

**Questions:**

- In 2.2, how come the example given is only relevant for safe RL with average safety cost constraints, and not for probability 1 constraints?
- How would you extend PI/Q-simmer for safety selection to more complex state and action spaces?
- It seems [37] also highlights the importance of the safety state. Can the authors clarify what is different about this work that makes the safety state seem more important?

**Limitations:**

Not really addressed.

**Strengths And Weaknesses:**

 ## Strengths

**Results:** Strong results in RL SafetyGym benchmark tasks

**Writing:** Explanations are generally good and intuitive, and writing is generally clear.

**Method:** An intuitive yet pretty simple method of selecting the safety constraint on-demand.

## Weaknesses

**Related Works:** Some missing safety papers taking different approaches the authors could cite and compare: [https://arxiv.org/pdf/1702.01182.pdf](https://arxiv.org/pdf/1702.01182.pdf), [https://arxiv.org/abs/2008.06622](https://arxiv.org/abs/2008.06622),  [https://www.aaai.org/AAAI22Papers/AAAI-5810.MaY.pdf](https://www.aaai.org/AAAI22Papers/AAAI-5810.MaY.pdf)

**Writing/Clarity:** One issue I have with the writing is in the presentation of the baselines. There are quite a few baselines and ablations (Simmer PID-L, PID-L, L-PPo, Lambda, PID-L w SA,… etc). The experiments section would be a lot easier to read and think about if each baseline was explicitly written out (bullet pointed list? with bolded names for each baseline and a quick description?) in some subsection.

Similarly, paragraph 3 of the introduction describes the method in a bit too much detail and is hard to follow when reading for the first time without prior knowledge about the method, safety states, etc.

Also, “SAUTE MDP” is never defined but just used in the paper (you cite [37] but don’t mention the word saute).

A custom environment is made, but there isn’t a picture of it in either the main paper or appendix, making it hard to think about and contextualize the results on it. It would also make sense to just have an overview figure of all environments (including some example safety gym ones) for clarity.

---

> ### Author Response · Authors · 2022-07-28
> **Response to weaknesses and general comments**
>
> **(Edited on 07/08)**
>
> We thank the reviewer for their careful consideration of our submission and their comments. Hopefully, our response will strengthen our paper and improve their opinion of our paper. Please see the general response, as well! We are looking forward to your reply!
>
> ## Weaknesses and general comments
> > Some missing safety papers taking different approaches the authors could cite and compare: https://arxiv.org/pdf/1702.01182.pdf, https://arxiv.org/abs/2008.06622, https://www.aaai.org/AAAI22Papers/AAAI-5810.MaY.pdf
>
> We cite and discuss the first two papers *in the revised version*. We *also* run CAP (AAAI paper) and a recent ICML publication (CVPO) on PointGoal1 and added the figures to appendix.
>
> We tried CAP with different planning horizons and different discount factors, but at the moment we cannot get returns in double digits. CVPO shows a good return, but the cost rates are not performing well and longer running time will not change this performance.  Furthermore, based on the logs it appears that there is problem with the E-step, since one of the dual variables ($\eta$) is taking the values close to zero. Overall, these results are too preliminary to include in the paper, but the figures are in Appendix for reviewers' convenience (Figures A6).
>
> > One issue I have with the writing is in the presentation of the baselines. There are quite a few baselines and ablations (Simmer PID-L, PID-L, L-PPo, Lambda, PID-L w SA,… etc). The experiments section would be a lot easier to read and think about if each baseline was explicitly written out (bullet pointed list? with bolded names for each baseline and a quick description?) in some subsection.
>
> Thank you for the comment, we indeed have a few baselines and such a formalization helped. We expanded section 3.1 to introduce the baselines. Lines 202-210
>
> > Similarly, paragraph 3 of the introduction describes the method in a bit too much detail and is hard to follow when reading for the first time without prior knowledge about the method, safety states, etc.
>
> Thank you for pointing this out. We modified the paragraph in lines 35-40 to read as follows:
>
> ```
> In this work, we aim to enhance model-free safe reinforcement learning by augmenting the state-space with one state encapsulating the safety information. This safety state is initialized with the safety budget. If during the deployment a safety cost is incurred, we subtract this cost from the safety state thus tracking the remaining safety budget. The safety constraint now is simply making sure that the safety state remains nonnegative. Therefore, the value of the safety state can serve as a measure of distance to the unsafe region.
> ```
>
> > Also, “SAUTE MDP” is never defined but just used in the paper (you cite [37]) but don’t mention the word saute).
>
> Note [37] - Sootla et al 2022. We thought that the terminology of Sootla et al 2022 would be somewhat confusing in our case. This is because in Sootla et al 2022 Saute RL means Safe RL with probability one constraints, and Saute MDP means an MDP with an additional safety state. We also consider an MDP with an additional safety state, but we consider average constraints. To avoid confusion we did not use Saute terminology in our paper.
>
> > A custom environment is made, but there isn’t a picture of it in either the main paper or appendix, making it hard to think about and contextualize the results on it. It would also make sense to just have an overview figure of all environments (including some example safety gym ones) for clarity.
>
> Thank you for pointing out this oversight. We added a description in the appendix on Page a5. Note that the environment is available on github (see [49] - Yang et al 2021). We also added a remark in lines 248-249 stating that the intuition for the custom-made environment is explained in Figure 1 -- our motivational example. Apologies for this oversight.

---

> > ### Comment · Reviewer_UM3D · 2022-08-05
> > **Response to Authors**
> >
> > Thanks for the quick response.
> >
> > Regarding certain points:
> >
> > > We cite and discuss the first two papers.
> >
> > You did not in the original revision (I just checked again), but you do now in the updated version, so I think you could've clarified that in the response better as it seems misleading.
> >
> > > Thank you for the comment, we indeed have a few baselines and such a formalization helped. We expanded section 3.1 to introduce the baselines. Lines 206-214
> >
> > This is indeed much better, thank you. It could be perhaps formatted into a bulleted list to look better, but that's up to the authors to decide.
> >
> > > Thank you for pointing this out. We modified the paragraph in lines 35-40 to read as follows:
> >
> > Sorry, I think I may have been unclear the first time. There is too much detail in the introduction, in my opinion, about your method. It makes it **harder** to read as the introduction should be about introducing the main contributions and perhaps some difficulties that your method addresses, not about exactly how the actual method works itself. I do agree the new paragraph is clearer than the old one, however.
> >
> > > Note [37] - Sootla et al 2022. We thought that the terminology of Sootla et al 2022 would be somewhat confusing in our case. This is because in Sootla et al 2022 Saute RL means Safe RL with probability one constraints, and Saute MDP means an MDP with an additional safety state. We also consider an MDP with an additional safety state, but we consider average constraints. To avoid confusion we did not use Saute terminology in our paper.
> >
> > I'm specifically referring to Line 131, where you say "Saute MDP," without clarifying what its referring to or defining it beforehand. (I tried searching for "SAUTE" anywhere else in the paper and it only appears here and in the references).
> >
> > > Thank you for pointing out this oversight. We added a description in the appendix on Page a5. Note that the environment is available on github (see [49] - Yang et al 2021). We also added a remark in lines 256-257 stating that the intuition for the custom-made environment is explained in Figure 1 -- our motivational example. Apologies for this oversight.
> >
> > This is indeed an improvement, thank you. However, it would be nicer and clearer to have it in the main paper. If you find a way to reduce the text in the paper (for example, by shrinking the introduction) then you should put it in the main paper. However it's not a big deal.
> >
> >
> >
> > I appreciate the changes you made in the other response comment too.
> >
> > One more issue I have with the new version is that I believe the writing clarity and grammar of the newly added text is worse than the original text; hopefully the authors can address that before the end of the rebuttal period as they modify the paper in response to my and other reviewers' rebuttal responses.
> >
> > I will also carefully read and consider other reviewers' reviews and your responses to those reviews before deciding whether to change my score, but rest assured I will be active during the remainder of this discussion period. Thanks for the responses.

---

> > > ### Author Response · Authors · 2022-08-07
> > > **response**
> > >
> > > Thank you for the follow-up and clarifications. We appreciate these suggestions and tried our best to follow them.
> > >
> > > >> We cite and discuss the first two papers.
> > >
> > > > You did not in the original revision (I just checked again), but you do now in the updated version, so I think you could've clarified that in the response better as it seems misleading.
> > >
> > > Apologies for the confusion. We didn't intend to mislead, we meant that we cited these references after you brought them up. We edited our initial response to reflect this.
> > >
> > > > Sorry, I think I may have been unclear the first time. There is too much detail in the introduction, in my opinion, about your method. It makes it harder to read as the introduction should be about introducing the main contributions and perhaps some difficulties that your method addresses, not about exactly how the actual method works itself. I do agree the new paragraph is clearer than the old one, however.
> > >
> > > We felt but that we can explain in plain language the main idea behind our approach, but it appears that our attempts were less successful than we thought. We removed a few sentences from this paragraph as the reviewer suggested.
> > >
> > > > I'm specifically referring to Line 131, where you say "Saute MDP," without clarifying what its referring to or defining it beforehand. (I tried searching for "SAUTE" anywhere else in the paper and it only appears here and in the references).
> > >
> > > Thank you for spotting this! Corrected.
> > >
> > > > This is indeed an improvement, thank you. However, it would be nicer and clearer to have it in the main paper. If you find a way to reduce the text in the paper (for example, by shrinking the introduction) then you should put it in the main paper. However it's not a big deal.
> > >
> > > We made some math more compact and shrunk the introduction (especially, the discussion on the curriculum learning approach and description of the safety state). This allowed fitting the figure partially (no pictures of car environments though).
> > >
> > > > One more issue I have with the new version is that I believe the writing clarity and grammar of the newly added text is worse than the original text;
> > >
> > > We made a few more passes on the newly added text and will make a few more in the coming days. Rest assured we will equalize the text quality for the final version. We will upload another version tomorrow.

---

> > > > ### Comment · Reviewer_UM3D · 2022-08-07
> > > > **Final Response**
> > > >
> > > > I thank the authors for being very active in this rebuttal period. After looking through the other reviews and authors' responses, and the updated manuscript, I believe this paper is near-ready for acceptance and that the authors may continue to slightly iterate on this paper during/after this review process.
> > > >
> > > > As such, i am changing my score to a 7.

---

> > > > > ### Author Response · Authors · 2022-08-07
> > > > > **thank you for a constructive review process!**
> > > > >
> > > > > and apologies again for misunderstandings and typos!

---

> > > > > ### Author Response · Authors · 2022-08-08
> > > > > **one minor change for the rebuttal**
> > > > >
> > > > > One of the reviewers was unhappy that we had 10 pages in our submission and we had to temporarily remove a figure with environments and 4 out of 6 figures with learning curves for the safety gym benchmark.
> > > > >
> > > > > Apologies for the last minute, but hopefully a temporary change.

---

> ### Author Response · Authors · 2022-07-28
> **Response to specific questions**
>
> **(Edited on 07/08)**
>
> We thank the reviewer for their careful consideration of our submission and their comments. Hopefully, our response will strengthen our paper and improve their opinion of our paper. Please see the general response, as well! We are looking forward to your reply!
>
> ## Questions
>
> > In 2.2, how come the example given is only relevant for safe RL with average safety cost constraints, and not for probability 1 constraints?
>
> Thank you for pointing this out. The logic of the example actually does not depend on the type of the constraints, we remove the reference to probability one constraints to avoid confusion.
>
> > How would you extend PI/Q-simmer for safety selection to more complex state and action spaces?
>
> We do not foresee any special technical difficulties in more complex state and action spaces since PI and Q Simmer algorithms receive only safety violation information. We agree, however, that a certain degree of adaptation may be required. In the case of multiple constraints, one could consider an aggregate violation of all constraints. We added a few sentences in the discussion in lines 318-321.
>
> > It seems [37] also highlights the importance of the safety state. Can the authors clarify what is different about this work that makes the safety state seem more important?
>
>  In our point of view, Sootla et al 2022 ([37]) presented the Safe RL algorithm with probability one constraints, for which they employed safety state augmentation. Our focus is on the broader impact of the safety state augmentation and its necessity for safe RL, in general. We consider both constraints on average and constraints with probability one. We further focus on other aspects of using the safety state and scheduling the safety budget (see points 1.-4.). We added some comments in the introduction to make things clearer. See lines 40-50
>
> Our main message is that safety state augmentation is vital for safe RL, whether the constraints are with probability one or on average. Our main contributions are as follows:
>
> 1. We present a novel problem formulation for safety during training. We apply this formulation to Safe RL with probability one constraints. We derive two algorithms for solving the problem (albeit with some limitations, e.g., an inevitable initial burst of constraint violations)
> 2. We show that for simple environments safety during training for safe RL with constraints on average is achieved without any other modification to Lagrangian PPO. We just need to add the safety state!
> 3. We offer state-of-the-art performance for Safe RL with constraints on average on safety gym benchmarks (Car and Point robots). We note that the safety gym benchmark is considered to be the most challenging set of environments for Safe RL.
> 4. Two environments (swing up pendulum and safety gym) where Lagrangian methods do not perform well in terms of safety but simply adding the safety state solves all the issues. We note that while the solution is simple, it is novel and the problem is very hard to solve without our approach.
>
> We added further experiments to support claim 3. We stress again that no other paper in the literature (to our best knowledge) has similar results. We believe that the points 1.-4. popularize **the need** for safety state augmentation. Note that the summary of Sootla et al 2022 is presented in Appendix.

---

### Official Review · Reviewer_KpTK · 2022-07-02

**Rating:** 3
**Confidence:** 4
**Soundness:** 3 good
**Presentation:** 1 poor
**Contribution:** 1 poor

**Summary:**

The paper considers the problem of safety reinforcement learning (model-free) in which the goal is to train an agent to not only maximize the average return but also satisfy an average safety violation cost along the trajectories. The approach of augmenting the state space plus shaping the reward in a 'saute' way (which can be applied to any RL algorithm) is being explored. The intuition behind the augmented state is the safety budget that is being diminished during the policy deployment, the paper introduces the SIMMER approach for scheduling the safety budget during the policy evaluation.

**Questions:**

* I do not get why the pure SAUTE method (without planning the budget) is not included in the baselines?
* Why the full suite of Safety-Gym benchmarks is not presented ?
* I think it is necessary to clearly show what are the advantages of the method except for marginal performance gains (w.r.t. average cost).
* Algorithms / introduced modifications should be presented in a standard way (using a pseudo-code) for better clarity.

## Minor remarks
* Eq. (4) the optimal control should be denoted by (e.g. $u^\star$),
* Eq. (6), (10) and others the symbols utilized are not defined formally,

**Limitations:**

Limitations and potential negative social impact of their work were properly addressed in the paper.

**Strengths And Weaknesses:**

The paper builds upon the SAUTE a state augmentation safety-RL method introduced in A. Sootla "SAUTE RL: Almost surely safe reinforcement learning using state augmentation" . The idea is based on switching from a constrained MDP into a standard MDP with augmented state space. The interpretation of the additional state variable is the safety budget depleted by each safety violation event.

The original SAUTE algorithm worked for a fixed safety budget.  The current paper as I understand proposes and evaluates an improved SAUTE by introducing two possible safety budget planning during deployment approaches. The resulting approach of SAUTE enhanced with safety budget planning is denoted by authors 'SIMMER'.  The two introduced methods for safety budget planning are based on 1. PI control, 2. Q-learning. The algorithm works for both settings - with the probability one safety constraint and the average constraints.

The paper is experimental the benchmarks include a safe Pendulum Swing-Up environment and the Safety-Gym benchmark suite. The baselines are: the Lagrangian PPO and Lagrangian TRPO methods introduced in the Safety-Gym paper, Constrained Policy Optimization (CPO), and PID Lagrangian from A Stooke et al, "Responsive safety in reinforcement learning by PID lagrangian methods".

 The approach is heavily based on the previously introduced method. It is not enough to fine-tune an existing approach, introduce a new trick, and coin a fancy-sounding acronym to obtain a new algorithm.

The Pendulum Swing-Up experiment shows an improvement over the baselines in terms of the safety constraint violations, over the Lagrangian PPO. However, this is a simple environment, and all of the methods reach a large task return pretty quickly.

In the Safety-Gym benchmarks restricted to the Point set of environments, an improvement is less clear. In cases where the average cost is less for SIMMER than those of the baseline PID Lagrangian method, the average return is significantly smaller for SIMMER. Only at the final stages of training (10 mln steps) does the average return of those methods equalize. For the PointGoal environment, it is visible that SIMMER has a lower variance than PID Lagrangian for the same average threshold at around 25. For me, it looks like it is a minuscule improvement over the PID Lagrangian approach. Appreciate that the PID Lagrangian baseline was carefully fine-tuned by the authors.

It is a pity that the probability one constraint setting was not examined in the safety environments (only the average cost setting is considered).

I do not understand why other benchmarks from Safety-Gym (Car and Doggo-based environments are not included), would show a more complete picture of the capabilities of the studied Safe-RL approaches. I just realized that 2 out of 3 total Car based environments are actually included in the appendix, and the results for Car-Push are looking much better than for Point-Push so I do not see what was author's motivation in making such a choice.

Conceptually the ideas are not well presented in the paper. I do not think that tabular form like (6) is clear for the reader. I understand that these are computational steps of an algorithm, so why not presenting them properly as an algorithm pseudo-code.

On the positive side, the experiments are well documented (ablation study and hyper-parameters are provided).

### Summary
I am not convinced that the introduced SIMMER algorithm is novel enough, and show a significant improvement over existing baselines to meet the publication bar on the NeurIPS conference.

---

> ### Author Response · Authors · 2022-07-28
> **Response to specific questions**
>
>
> We thank the reviewer for their careful consideration of our submission and their comments. Hopefully, our response would strengthen our paper and improve their opinion of our paper. Please see the general response as well!
>
> ## Questions
>
> > I do not get why the pure SAUTE method (without planning the budget) is not included in the baselines?
>
> Apologies for the confusion. PO PPO in Figure 2 is the Saute algorithm from Sootla et al 2022 ([37]). We modified the text to make this clearer (Lines 203-210).
>
> > Why the full suite of Safety-Gym benchmarks is not presented ?
>
> We believe our experiments are sufficient to illustrate the strengths and the weaknesses of our approach. We also present one of the most complete evaluations on the safety gym environments (6 out of 9) for recent safe RL papers, and our results are the best reported (to our best knowledge).  If the reviewer is aware of references negating (part of) this claim, we are very happy to cite them and add them to our evaluation.
>
> > I think it is necessary to clearly show what are the advantages of the method except for marginal performance gains (w.r.t. average cost).
>
> In our point of view, the gains are significant. We base this opinion on the following observations. In a simple custom-made safety gym environment PID Lagrangian is not performing well at all (Figure 4). The same can be said for safety metrics (cost rate, safety costs) in CarPush, PointPush environments (Figures 5 and 6). **Additionally, our goal was to improve the safety of the algorithms, which our results illustrate.**
>
> > Algorithms / introduced modifications should be presented in a standard way (using a pseudo-code) for better clarity.
>
> Thank you for this suggestion! We thought this form would be clearer, but we added the pseudo-code as the reviewer suggested and it does look better.
>
> **Finally, we kindly ask the reviewer to provide further technical flaws, instances of weak evaluation or bad reproducibility practices in order to improve our paper, if there are any left.**

---

> ### Author Response · Authors · 2022-07-28
> **Response to weaknesses (cont)**
>
>
> ## Weaknesses (cont)
> > In the Safety-Gym benchmarks restricted to the Point set of environments, an improvement is less clear. In cases where the average cost is less for SIMMER than those of the baseline PID Lagrangian method, the average return is significantly smaller for SIMMER. Only at the final stages of training (10 mln steps) does the average return of those methods equalize. For the PointGoal environment, it is visible that SIMMER has a lower variance than PID Lagrangian for the same average threshold at around 25. For me, it looks like it is a minuscule improvement over the PID Lagrangian approach. Appreciate that the PID Lagrangian baseline was carefully fine-tuned by the authors
>
> We will take these separately:
>
> >> In cases where the average cost is less for SIMMER than those of the baseline PID Lagrangian method, the average return is significantly smaller for SIMMER.
>
> In our point of view, the return should be evaluated at the end of training when the cost limits (safety budgets) are equal. Furthermore, the average cost for PID-Lagrangian is much noisier, while SIMMER keeps the cost down for the most part of the training and delivers the same return in the end. SIMMER also significantly outperforms PID Lagrangian in terms of the cost rate.
>
> >> Only at the final stages of training (10 mln steps) does the average return of those methods equalize.
>
> This is because we set out to achieve a good safety performance during training and return after training. We devised a schedule over 10 mln steps. Mainly we aimed to converge for every fixed safety budget and hence gave some time to converge. Naturally, PID Lagrangian does not have this limitation, but the safety violations are larger. As we mention to achieve safety we had to sacrifice sample efficiency.
>
> >> For the PointGoal environment, it is visible that SIMMER has a lower variance than PID Lagrangian for the same average threshold at around 25. For me, it looks like it is a minuscule improvement over the PID Lagrangian approach.
>
> In our point of view, PointGoal is a rather simple environment and we achieve better performance in other environments. In fact, our algorithm outperforms PID Lagrangian in terms of safety measures on all the environments. This is at the same time when the returns are fairly equal at the end of training. Furthermore, these are the best (to our knowledge) reported results on these environments.
>
> > I do not understand why other benchmarks from Safety-Gym (Car and Doggo-based environments are not included), would show a more complete picture of the capabilities of the studied Safe-RL approaches. I just realized that 2 out of 3 total Car based environments are actually included in the appendix, and the results for Car-Push are looking much better than for Point-Push so I do not see what was author's motivation in making such a choice.
>
> Thank you for this suggestion. We further tuned Car and Point environments (Point, Button and Push tasks) and added these results in the main text. We agree that adding 3 more environments would provide a better picture, but we believe our experiments illustrate our main claims well. Further, to our best knowledge no recent paper has such a complete set of experiments on the safety gym benchmark without making any modifications to the benchmark. If the reviewer is aware of such a paper we are very happy to cite them and add it to our evaluation.
>
> > Conceptually the ideas are not well presented in the paper. I do not think that tabular form like (6) is clear for the reader. I understand that these are computational steps of an algorithm, so why not presenting them properly as an algorithm pseudo-code.
>
> Thank you for this suggestion! We added the pseudo-code as the reviewer suggested and it does look better.

---

> ### Author Response · Authors · 2022-07-28
> **Response to general comments and weaknesses**
>
>
> We thank the reviewer for their careful consideration of our submission and their comments. Hopefully, our response would strengthen our paper and improve their opinion of our paper. Please see the general response as well!
>
> ## Summary
>
> > The paper considers the problem of safety reinforcement learning (model-free) in which the goal is to train an agent to not only maximize the average return but also satisfy an average safety violation cost along the trajectories. The approach of augmenting the state space plus shaping the reward in a 'saute' way (which can be applied to any RL algorithm) is being explored.
>
> We think there may be a slight misunderstanding. We consider two settings: constraints with probability one and constraints on average. In the average constrained case, we do not use reward shaping, the reward shaping is used only in the probability one constraint case.
>
> ## Weakenesses and general comments
>
> > The Pendulum Swing-Up experiment shows an improvement over the baselines in terms of the safety constraint violations, over the Lagrangian PPO. However, this is a simple environment, and all of the methods reach a large task return pretty quickly.
>
> Our interpretation of the results appears to differ from the reviewer's. Let us elaborate on why these experiments are important from our point of view.
>
> While we agree that the pendulum swing-up is a rather simple environment. We note, however, that comparable methods such as Turchetta et al 2020 ([42] in the first submission, now [46]) considered even simpler environments such as lunar lander (a simple stabilization environment) and frozen lake (finite state MDP), and an even simpler setting as they didn't learn online.
>
> We further stress that Simmer PPO not only outperforms Lagrangian PPO but also solves the task without any constraints violations – a result previously not reported by any model-free algorithm. This is achieved by adding a safety state to the environment. While the solution is simple, the problem was very hard to solve (in our opinion)
>
> > It is a pity that the probability one constraint setting was not examined in the safety environments (only the average cost setting is considered).
>
> We have tried probability one constraints with the PointGoal environment, but the results were not very convincing. After $10$ million steps and $3$ seeds, the returns were on average $2.44$, while the costs were $11.85$, the $90\%$-th percent quantile of the costs was equal to $18.00$. This means that the vast majority of trajectories were below the limit of $25$, but the returns were much lower than in the average constrained RL.
>
> We believe there are several things at play:
> 1. the PointGoal environment was designed with safety on average in mind and hence safety with probability one is hard to achieve,
> 2. The reward shaping aspect of the PO safe algorithm (Sootla et al 2022) may not be the best solution for this particular set of environments as it biases the reward, which is normally around zero.
>
> While it would be an interesting investigation in future, it lies outside of the scope of this paper. We also think this would be misleading at this point to present this comparison as we do not know the exact reasons for such behavior.
>
> > The approach is heavily based on the previously introduced method. It is not enough to fine-tune an existing approach, introduce a new trick, and coin a fancy-sounding acronym to obtain a new algorithm.
>
> We thank the reviewer for their opinion. Our main contributions are as follows:
>
> 1. We present a novel problem formulation for safety during training. We apply this formulation to Safe RL with probability one constraints. We derive two algorithms for solving the problem (albeit with some limitations, e.g., the inevitable initial burst of constraint violations)
> 2. We show that for simple environments safety during training for safe RL with constraints on average is achieved without any other modification to Lagrangian PPO. We just need to add the safety state!
> 3. We offer state-of-the-art performance for Safe RL with constraints on average on safety gym benchmarks (Car and Point robots). We note that the safety gym benchmark is considered to be the most challenging set of environments for Safe RL.
> 4. Two environments (swing up pendulum and safety gym) where Lagrangian methods do not perform well in terms of safety but simply adding the safety state solves all the issues. We note that while the solution is simple, it is novel and the problem is very hard to solve without our approach.
>
> We added further experiments to support claim 3. We stress again that no other paper in the literature (to our best knowledge) has similar results. We believe that the points 1.-4. popularize the need for safety state augmentation.

---

> ### Author Response · Authors · 2022-07-28
> **Table with numerical comparsion of results**
>
> For the ease of comparison of performances, we compiled two tables of numerical values of the results at the end of training. We compute the statistics over the last $15$ epochs and present mean $\pm$ standard deviation. We observe that the cost rates for Simmer are significantly smaller than the cost rates for PID Lagrangian, on CarButton1 and CarPush1 Simmer PID Lagrangian outperforms PID Lagrangian in terms of the returns as well. These tables is added to the appendix.
>
> Simmer PID Lagrangian
>
> | Environment     | Return              | Cost            | Cost rate ($\cdot 1e^2$)
> |-----------------|---------------------|-----------------|-------------------------|
> | PointGoal1-v0   | $25.24\pm 0.81$     | $24.64\pm 2.72$ | $\bf{1.77\pm 0.01}$    |
> | PointButton1-v0 | $10.36\pm 0.93$     | $24.37\pm 2.81$ | $\bf{1.83\pm 0.01}$  |
> | PointPush1-v0   | $10.27\pm 3.45$     | $25.93\pm 3.58$ | $\bf{1.77\pm 0.06}$     |
> | CarGoal1-v0     | $23.97\pm 2.43$     | $25.31\pm 2.67$ | $\bf{1.78\pm 0.02}$     |
> | CarButton1-v0   | $\bf{1.93\pm 0.64}$ | $25.95\pm 5.39$ | $\bf{1.81\pm 0.01}$     |
> | CarPush1-v0     | $\bf{8.01\pm 0.64}$ | $17.60\pm5.48$  | $\bf{1.44\pm0.05}$      |
>
> PID - Lagrangian
> | Environment    | Return               | Cost            | Cost rate ($\cdot 1e^2$) |
> |-----------------|----------------------|-----------------|--------------------------|
> | PointGoal1-v0    | $26.23\pm 0.22$      | $25.55\pm 3.62$ | $2.51\pm 0.00$           |
> | PointButton1-v0   | $10.50\pm 1.10$      | $23.98\pm 3.73$ | $2.56\pm 0.01$           |
> | PointPush1-v0     | $10.50\pm 4.41$      | $23.55\pm 4.59$ | $2.28\pm 0.26$           |
> | CarGoal1-v0        | $\bf{27.46\pm 2.35}$ | $22.68\pm 3.77$ | $2.55\pm 0.01$           |
> | CarButton1-v0    | $1.09\pm 0.83$       | $24.98\pm 6.53$ | $2.55\pm 0.01$           |
> | CarPush1-v0       | $6.74\pm 1.11$       | $19.41\pm6.10$  | $2.01\pm 0.19$           |

---

### Official Review · Reviewer_vED7 · 2022-07-09

**Rating:** 7
**Confidence:** 5
**Soundness:** 3 good
**Presentation:** 3 good
**Contribution:** 3 good

**Summary:**

This paper presented a novel state augmentation technique to enhance safe exploration of the RL problem with safety budget constraints. Both probability-one constraint and average constraint are considered in the problem formulation. A safety budget schedule is designed to reduce constraint violations and guide safe exploration in the simmering safe RL algorithm. The experimental results well supported the claimed contributions. I will consider further improving my rating if the author could explain the connection and difference between this paper and the reference [37].

**Questions:**

1. The major question I am interested in is what is the difference between this paper and  [37]?

   Other minor questions about technical details:

2. For the probability one constraints, the notion $P(z\geq 0) - 1 \leq d$ is confusing because the left-hand side must be lower than zero. However, the budget is always positive in the experiment section. Could you explain that?

3. Can the proposed methods handle safety constraints other than the safety budget?

**Limitations:**

The paper title starts with *enhancing safe exploration*, which might be too strong for the contemporary model-free safe RL algorithms. I agree that the proposed algorithm has significantly improved safety during training with a solid and novel solution. However, many other challenges still remain unfixed when referring to the **safe exploration** problem, such as violation bursts in the initial training stages, violations caused by variance and disturbances during training, lack of provable safety guarantees, etc. I suggest the author explain more the contributions and limits of this paper in the context of the safe exploration problem.

**Strengths And Weaknesses:**

Strength

1. This paper is well-written, and the solution is novel and clearly presented.
2. The experimental results well supported the claimed contribution. The contribution of safety enhancement is solid.

Weakness

1. This motivation of this paper is similar to the paper [37] in the reference. The author has not thoroughly explained the connection and difference between this paper and [37].

---

> ### Author Response · Authors · 2022-07-28
> **Response to a comment on limitations**
>
> ## Limitations
>
> > The paper title starts with enhancing safe exploration, which might be too strong for the contemporary model-free safe RL algorithms. I agree that the proposed algorithm has significantly improved safety during training with a solid and novel solution. However, many other challenges still remain unfixed when referring to the safe exploration problem, such as violation bursts in the initial training stages, violations caused by variance and disturbances during training, lack of provable safety guarantees, etc. I suggest the author explain more the contributions and limits of this paper in the context of the safe exploration problem.
>
> Thank you for this comment. We changed the title to ``Effects of Safety State Augmentation on Safe Exploration''. Indeed, there are many limitations and future work directions that we failed to mention. We also note that it appears that the initial safety budget has an effect on the initial burst of constraint violations. We use this feature to lower the number of constraint violations with respect to the target safety budget. We added a discussion on limitations to reflect the reviewer's comments (lines 315-326).
>
> ```
> Augmenting the safety state and scheduling the safety budget do not solve all the problems in safe exploration. First, an initial burst of constraint violations is an inevitable reality of using our model-free approach. It appears that lowering the initial safety budget lowers the initial burst of constraint violations, but does not completely solve the problem. We do not see how to address this limitation without making further assumptions on the environment. Second, it is noticeable that the performance on the PointPush1 environment is quite noisy for both PID-L and Simmer PID-L. This is because some seeds achieve very good performance (return of 15) and some seeds do not (return of 5), while the learning curves appear to be stable. This suggests that the learning procedure is at a local maximum. In our point of view, this calls for a more sophisticated algorithm for exploration seeking maximal return. This may help achieve stable performance in this environment. Beyond these limitations, one can also list safety violations caused by variance and disturbances, lack of provable safety guarantees, etc. Studying these limitations is outside of the scope of this paper.
> ```

---

> ### Author Response · Authors · 2022-07-28
> **Response to weaknesses and specific comments**
>
> We thank the reviewer for their careful consideration of our submission and their comments. Hopefully, our response would strengthen our paper and improve their opinion of our paper. Please see the general response as well!
>
> ## Weaknesses
> > The major question I am interested in is what is the difference between this paper and [37]?
>
> In our point of view, Sootla et al 2022 ([37]) presented the Safe RL algorithm with probability one constraints, for which they employed safety state augmentation. Our focus is on the broader impact of the safety state augmentation and its necessity for safe RL, in general. We consider both constraints on average and constraints with probability one. We further focus on other aspects of using the safety state and scheduling the safety budget (see points 1.-4.). We added some comments in the introduction to make things clearer. See lines 40-53
>
> Our main message is that safety state augmentation is vital for safe RL, whether the constraints are with probability one or on average. Our main contributions are as follows:
>
> 1. We present a novel problem formulation for safety during training. We apply this formulation to Safe RL with probability one constraints. We derive two algorithms for solving the problem (albeit with some limitations, e.g., an inevitable initial burst of constraint violations)
> 2. We show that for simple environments safety during training for safe RL with constraints on average is achieved without any other modification to Lagrangian PPO. We just need to add the safety state!
> 3. We offer state-of-the-art performance for Safe RL with constraints on average on safety gym benchmarks (Car and Point robots). We note that the safety gym benchmark is considered to be the most challenging set of environments for Safe RL.
> 4. Two environments (swing up pendulum and safety gym) where Lagrangian methods do not perform well in terms of safety but simply adding the safety state solves all the issues. We note that while the solution is simple, it is novel and the problem is very hard to solve without our approach.
>
> We added further experiments to support claim 3. We stress again that no other paper in the literature (to our best knowledge) has similar results. We believe that the points 1.-4. popularize **the need** for safety state augmentation.
> Note that the summary of Sootla et al 2022 is presented in Appendix.
>
> We hope this clarifies the difference
>
> ## Questions
> > For the probability one constraints, the notion
> $$P(z\ge 0)−1\le d$$
> > is confusing because the left-hand side must be lower than zero. However, the budget is always positive in the experiment section. Could you explain that?
>
> Apologies for the confusion, but the way we wrote it the constraint actually reads
> $$P(z\ge 0)−1 \ge 0,$$
> implying that the probability should be greater or equal to 1, which means "with probability equal to one". We acknowledge that this is confusing, but we couldn’t find a less confusing way of writing this.
>
> > Can the proposed methods handle safety constraints other than the safety budget?
>
> The main idea of the safety state is to replace an accumulated constraint with a hard constraint, e.g., $z_t \ge 0$ for all $t$. In light of this comment, using the safety state for hard constraints could be redundant. Perhaps, different forms of safety states could be considered such as Euclidean distance toward hazards. However, we are not certain that these changes will be crucial for learning.

---

> > ### Comment · Reviewer_vED7 · 2022-08-07
> > **Reply to the author rebuttal**
> >
> > Thank the authors for kindly addressing my concerns!
> >
> > First, for the main problem and question I mentioned, the connection between the submitted paper and reference [37] in the original submission, I think the differences are clear and novel enough. Especially, I like the budget scheduling part proposed in the submitted paper. It is a simple but effective approach to some difficulties in practical constrained RL. Furthermore, this approach might be related to some curriculum learning studies, it would be interesting to dig deeper into the intersections of these fields.
> >
> > I am also happy to see the authors pay attention to the limitations (I will summarize the replies here). To be honest, safe RL, especially the model-free one, is still an open problem. Therefore, it is critical to adequately state the contributions and limitations of a safe RL study.
> >
> > Overall, This paper presented an interesting and effective method for safe RL. The experimental results well support the claimed contribution. I think this paper is meaningful for the safe/constrained RL community (as well as the intersection with curriculum learning possibly). The authors also answered all my questions in the rebuttal. I do not have further questions, and I decide to change my rating to 7.

---

> > > ### Author Response · Authors · 2022-08-07
> > > **Thank you for a great review!**
> > >
> > > We thank the reviewer again for their time and their comments, which lead to the improvement of our submission.

---

### Official Review · Reviewer_n7QX · 2022-07-10

**Rating:** 4
**Confidence:** 4
**Soundness:** 3 good
**Presentation:** 3 good
**Contribution:** 2 fair

**Summary:**

The authors present a method for safe RL by augmenting the original MDP with an additional state which functionally serves as a "distance" to constraint violation. A method is presented to learn the initial value of the cost function (referred to a as a safety budget). The authors study the case where the constraints are imposed on the average cost, and where the constraints are imposed on the cost with probability 1. The authors compare their approach against some benchmarks and show reduced constraint violations.

**Questions:**

Have the authors considered the use of temporal logics and synthesis in order to specify the safety requirement? For example the following paper: (Jiang, Y., Bharadwaj, S., Wu, B., Shah, R., Topcu, U., & Stone, P. (2021). Temporal-Logic-Based Reward Shaping for Continuing Reinforcement Learning Tasks).

These methods, in essence results in an augmented MDP with additional safety states. While the linked paper uses the "distance to the safety state" as a reward shaping function, a separate cost function may be applied instead.

Is it correct to assume that assigning a cost function to these states will result in the same outcome as the work presented in this paper? If not, perhaps a comparison would help strengthen the authors' contribution.

**Strengths And Weaknesses:**

The paper is presented well and explained. However, the writing could be further enhanced with the use of toy examples to illustrate concepts.

It seems one of the main contributions of this paper is learning the safety budget. There are certain aspects regarding the tuning of the safety budget that seems ad-hoc or could do with further explanations and concrete testing. For example, the authors state: "If the current accumulated costs are well above the safety budget then the safety budget should be decreased to ensure that the policy is incentivized to be safer". It is not clear why decreasing the safety budget would help here. If the policy is already causing significant safety violations above the safety budget, it seems that the safety requirement just may not feasible in that setting?

The authors compare against related algorithms in RL, but there are several different approaches to safe RL that could still be compared against for more empirical evidence. For example the paper "Safe Reinforcement Learning by Probabilistic Shields" is another approach to safe RL, but allows the user to specify a probability of safety violation. In contrast the safety budget approach doesn't have a direct relationship on the amount of violation a user is able to accept.

Overall, while this paper does present an interesting idea, it seems to be a somewhat incremental contribution. It relies on a lot of tuning and doesn't compare enough against other safe RL approaches.

---

> ### Author Response · Authors · 2022-07-28
> **Response to questions**
>
> We thank the reviewer for their careful consideration of our submission and their comments. Hopefully, our response would strengthen our paper and improve their opinion of our paper. Please see the general response as well! We would kindly request the reviewer to suggest further steps to improve our paper since we have addressed all the previous reviewer's comments from our point of view.
>
> ## Question
>
> > Have the authors considered the use of temporal logics and synthesis in order to specify the safety requirement? For example the following paper: (Jiang, Y., Bharadwaj, S., Wu, B., Shah, R., Topcu, U., & Stone, P. (2021). Temporal-Logic-Based Reward Shaping for Continuing Reinforcement Learning Tasks).
>
> > These methods, in essence results in an augmented MDP with additional safety states. While the linked paper uses the "distance to the safety state" as a reward shaping function, a separate cost function may be applied instead.
>
> > Is it correct to assume that assigning a cost function to these states will result in the same outcome as the work presented in this paper? If not, perhaps a comparison would help strengthen the authors' contribution.
>
> To our best understanding, Jiang et al 2021 solve a reward shaping problem, not safety. Considering we are not very familiar with temporal logics, we cannot answer the reviewer's question and it seems to require a deeper investigation. The question does seem interesting and warrants an investigation in future work.
>
> We thank the reviewer for bringing up Jiang et al 2021 and Jansen 2020, but we do not see how to perform a direct comparison to them.

---

> ### Author Response · Authors · 2022-07-28
> **Response to general comments and weaknesses**
>
> **(Edited on 07/08)**
>
> We thank the reviewer for their careful consideration of our submission and their comments. Hopefully, our response would strengthen our paper and improve their opinion of our paper. Please see the general response as well! We would kindly request the reviewer to suggest further steps to improve our paper since we have addressed all the previous reviewer's comments from our point of view.
>
> > The paper is presented well and explained. However, the writing could be further enhanced with the use of toy examples to illustrate concepts.
>
> Thank you for pointing this out. We improved the section on PI Simmer by stripping the algorithm to its main bits and providing an intuition (lines 166-179). We already use toy examples to explain the need for the safety state augmentation and actually confirm this logic on the custom-made safety gym environment (lines 248-249).
>
> > It seems one of the main contributions of this paper is learning the safety budget. There are certain aspects regarding the tuning of the safety budget that seems ad-hoc or could do with further explanations and concrete testing.
>
> We kindly ask the reviewer for some suggestions on testing and points that require clarification.
>
> > For example, the authors state: "If the current accumulated costs are well above the safety budget then the safety budget should be decreased to ensure that the policy is incentivized to be safer". It is not clear why decreasing the safety budget would help here. If the policy is already causing significant safety violations above the safety budget, it seems that the safety requirement just may not feasible in that setting?
>
> Overall, our intuition relies on a series of empirical observations. In particular, we have observed that reasonably low safety budgets result in lower bursts of constraint violations during training. Note also that for average constrained problems we use a simple schedule without adjustments during training. We also assume that policies with different safety budgets are feasible (lines 127-132), and hence this problem should not occur. We add a discussion in Lines 159-165:
>
> ```
> To alleviate this issue we propose two approaches: PI Simmer (PI-controlled safety budget) and Q Simmer (Online Q-learning with non-stationary rewards). In both cases the intuition is similar. If the current accumulated costs are well below the safety budget $\bmd_k^{\rm ref}$, then we are very safe and the safety budget can be further increased. If the accumulated costs are around the safety budget, then we could stay at the same level or increase the safety budget. If the current accumulated costs are well above the safety budget $\bmd_k^{\rm ref}$, then the safety budget should be decreased. This intuition is based on our previous empirical observations, i.e., lower safety budgets would often cause lower initial bursts in accumulated safety costs. We hypothesize that the policy can quickly learn ``extremely unsafe'' actions thus providing low safety cost bursts for low safety budgets.
> ```
>
> > The authors compare against related algorithms in RL, but there are several different approaches to safe RL that could still be compared against for more empirical evidence.
>
> We focused on improving a particular set of algorithms using safety state augmentation and we believe comparing to this set of algorithms is a fair assessment. *Nevertheless we run two more baselines on PointGoal1*
>
> > For example the paper "Safe Reinforcement Learning by Probabilistic Shields" is another approach to safe RL, but allows the user to specify a probability of safety violation. In contrast the safety budget approach doesn't have a direct relationship on the amount of violation a user is able to accept.
>
> As far as we can see Jansen 2020 consider a finite state MDP and cannot be directly compared on the safety gym benchmarks. While we cite Jansen 2020, we refrain from a direct comparison. However, we welcome further suggestions and we are happy to run appropriate baselines. We are running some baselines at the moment.
>
> > Overall, while this paper does present an interesting idea, it seems to be a somewhat incremental contribution. It relies on a lot of tuning and doesn't compare enough against other safe RL approaches.
>
> We compare against 5 baselines, we added *two further baselines* and *a few further environments* in the rebuttal (now we have 8 environments). We are happy to add further baselines, but none of the suggested baselines actually use the Safety Gym benchmarks making such a comparison hard. We note that the safety gym benchmark is considered to be the most challenging set of environments for Safe RL.

---

> > ### Comment · Reviewer_n7QX · 2022-08-07
> > **Response to authors**
> >
> > I appreciate your response. My main concern however is that the work here relies on a lot of tuning based on empirical observations as you state in your response. This makes it challenging for reproducibility and generalizability outside of benchmark problems.
> >
> > The authors have put in a lot of effort into comparing against baselines and this is important and necessary work. However, I think another important facet is to make a theoretical case on why the proposed method should outperform other methods. Or atleast, the authors should make an argument on what types of scenarios is the algorithms the authors present theoretically superior to the state of the art.

---

> > > ### Author Response · Authors · 2022-08-07
> > > **a clarification**
> > >
> > > **Edited**
> > >
> > > We thank the reviewer for the reply.  We agree that there may be different opinions on novelty, significance, and performance, but we believe here the problem is a misunderstanding, which we hope we can clarify.
> > >
> > > >  My main concern however is that the work here relies on a lot of tuning based on empirical observations as you state in your response.
> > >
> > > We think there's a slight misunderstanding of our work. We can argue that our algorithms depend on tuning less than the baselines since the behavior of our learning curves is generally more stable. Further, *our intuition* is based on empirical observations, but the algorithms are justified with the same level of rigor as our baselines.
> > >
> > > > This makes it challenging for reproducibility and generalizability outside of benchmark problems.
> > >
> > > Reproducibility and generalizability are very serious, but common problems in RL. We kindly ask the reviewer to specify in what aspect our algorithm is less reproducible than our baselines (PID - Lagrangian, Lambda, Lagrangian PPO, CPO) and we hope we can address this in future work. We note, however, that we can "Simmer" any of those baselines, but if these baselines are not suitable, we are open to considering other baselines.
> > >
> > > > The authors have put in a lot of effort into comparing against baselines and this is important and necessary work.
> > >
> > > Thank you for your kind words, but we actually put a moderate amount of effort into tuning. We didn't get drastic performance changes during our tuning efforts.
> > >
> > > > However, I think another important facet is to make a theoretical case on why the proposed method should outperform other methods. Or at least, the authors should make an argument on what types of scenarios is the algorithms the authors present theoretically superior to the state of the art.
> > >
> > > We think there is a slight misunderstanding of our work. We justify the use of the safety state in Section 2.2 and cite relevant theoretical work supporting our claims. In our opinion, these observations alone justify our approach to safe RL in comparison to state-of-the-art.
> > >
> > > We consider two settings: constraints with probability one and constraints on average.  In the probability one constraint setting there are really no comparable works (Sootla et al 2022 do not consider the problem of training safely). In the average constraint case, our algorithm outperforms all the baselines in all benchmark problems on all the metrics. The superior performance in both cases is shown to be due to the safety state augmentation.
> > >
> > > We are happy to give a more detailed response if the reviewer can offer more specifics regarding the drawbacks of our approach.

---

### Author Response · Authors · 2022-07-28
**General response to all reviewers**

**(Edited on 07/08)**

We want to thank all the reviewers for the time and evaluation of our work. We wish we could convey the message in a better way and regret several misunderstandings.

## Issues raised by several reviewers

**Difference to Sootla et al 2022 ([37] in the initial submission).**  In our point of view, Sootla et al 2022 ([37]) presented the Safe RL algorithm with probability one constraints, for which they employed safety state augmentation. Our focus is on the broader impact of the safety state augmentation and its necessity for safe RL, in general. We consider both constraints on average and constraints with probability one. We further focus on other aspects of using the safety state and scheduling the safety budget (see points 1.-4.). We added some comments in the introduction to make things clearer. See lines 40-50

Our main message is that safety state augmentation is vital for safe RL, whether the constraints are with probability one or on average. Our main contributions are as follows:

1. We present a novel problem formulation for safety during training. We apply this formulation to Safe RL with probability one constraints. We derive two algorithms for solving the problem (albeit with some limitations, e.g., an inevitable initial burst of constraint violations)
2. We show that for simple environments safety during training for safe RL with constraints on average is achieved without any other modification to Lagrangian PPO. We just need to add the safety state!
3. We offer state-of-the-art performance for Safe RL with constraints on average on safety gym benchmarks (Car and Point robots). We note that the safety gym benchmark is considered to be the most challenging set of environments for Safe RL.
4. Two environments (swing up pendulum and safety gym) where Lagrangian methods do not perform well in terms of safety but simply adding the safety state solves all the issues. We note that while the solution is simple, it is novel and the problem is very hard to solve without our approach.

We added further experiments to support claim 3. We stress again that no other paper in the literature (to our best knowledge) has similar results. We believe that the points 1.-4. popularize **the need** for safety state augmentation. Note that the summary of Sootla et al 2022 is presented in Appendix.

**Further baselines.** Our baselines were tuned in the past on the used environments or we have spent a significant amount of time tuning them. We note that there are not many papers in the literature which have actually used Safety Gym benchmark environments as is without modifications. These Safety Gym benchmarks are considered to be the hardest benchmarks for Safe RL. Nevertheless, we will try to add further baselines that reviewers suggested and we ask for more suggestions.

## Major changes at this moment:
1. Added new Car and Push experiments to the main file tuning the performance
2. Added an algorithm description for PI Simmer and Q Simmer
3. Simplified and improved the presentation of PI SIMMER
4. Added a paragraph discussing the limitations of our techniques.
5. Changed the title to ``Effects of Safety State Augmentation on Safe Exploration''
6. We run 2 baselines: CAP [Ma et al 2022] and CVPO [Liu et al 2022] (see Figure A6 in Appendix), and they underperform. It is not clear if it's due to tuning or the algorithms themselves, but we've used default parameters with limited tuning.
7. We cut the introduction, and added figures with environments as was requested by reviewer UM3D.

Note that:
1. Due to added references, some reference numbers are different in a new submission. E.g., [37] -> [41]
2. We mark changes in magenta color.
3. We apologize for possible typos.
4. We tried to keep the line references correct, but apologies if some don't match with the text.

[Ma et al 2022] Ma, Yecheng Jason, et al. "Conservative and adaptive penalty for model-based safe reinforcement learning." Proceedings of the AAAI Conference on Artificial Intelligence. Vol. 36. No. 5. 2022.

[Liu et al 2022] Liu, Z., Cen, Z., Isenbaev, V., Liu, W., Wu, S., Li, B., & Zhao, D. (2022, June). Constrained variational policy optimization for safe reinforcement learning. In International Conference on Machine Learning (pp. 13644-13668). PMLR.

---

> ### Comment · Reviewer_KpTK · 2022-08-08
> **final remarks**
>
> Thank you for the additional work. I read the revised version,  other reviews, and rebuttals.
> For the moment, I consider my review final. I have not changed my judgment about the paper.
> Before I reiterate my concerns, I like to mention that this is promising work, safety gym environments are hard,
> and the work can be more impactful when presented appropriately.
> My main concerns, which still hold, are as follows:
>
> * the experimental improvement over existing baseline methods on which the presented method is based is unclear. When viewing the problem as maximization of the reward by keeping the average cost below a safety threshold, the improvement in cost rate exclusively is not as impressive. Maybe restating the problem and decreasing the safety threshold would show clearer improvements over the baselines? Researchers who ate their teeth in a safety-gym may find even the slight improvements in cost values important, but for the general audience, it is harder to notice the method's advantages.
>
> * The methodological novelty of the approach is limited, e.g., compared to existing work Sootla et al. 2022 'SAUTE' , I am not convinced that the SIMMER has a significant degree of novelty. It seems to be a slight implementation improvement with the cost budget simmering and evaluated in the average cost environment setting, which is a natural step after solving the prob. One safety setting using the original approach.
>
> *  I agree with Reviewer n7QX that the improvements resulting from empirical fine tuning on the given set of environments may be hard to generalize to a wider set of benchmarks. Hence it is important to give a theoretical case on why the method may outperform other approaches.
>
> * The current revised version of the paper differs significantly from the originally submitted version and should undergo a new review process. E.g., a new set of benchmarks appeared in the main text after further 'fine-tunning' was done by the authors.
>
> * The current revision is in excess of 9 pages, which violates the revision rules (the 9-page limit still holds for the revision). Hence I have not analyzed all of the new content carefully.
>
> * the clarity of the presentation should be improved and made more formal, e.g., the weird tables with the algorithm steps.

---

> > ### Author Response · Authors · 2022-08-08
> > **response to final remarks**
> >
> > We thank the reviewer for their response and the kind words.
> >
> > > the experimental improvement over existing baseline methods on which the presented method is based is unclear. When viewing the problem as maximization of the reward by keeping the average cost below a safety threshold, the improvement in cost rate exclusively is not as impressive. Maybe restating the problem and decreasing the safety threshold would show clearer improvements over the baselines? Researchers who ate their teeth in a safety-gym may find even the slight improvements in cost values important, but for the general audience, it is harder to notice the method's advantages.
> >
> > One of our main points is that the cost rate is a measure of safety during training. Improving this measure was the goal of the paper. We can agree to disagree if this is important. We again point out that there are further contributions. We can agree to disagree if they are significant, but we shouldn't disregard them.
> >
> > > The methodological novelty of the approach is limited, e.g., compared to existing work Sootla et al. 2022 'SAUTE' , I am not convinced that the SIMMER has a significant degree of novelty. It seems to be a slight implementation improvement with the cost budget simmering and evaluated in the average cost environment setting, which is a natural step after solving the prob. One safety setting using the original approach.
> >
> > Again we can agree to disagree about novelty, but it's confusing to claim that a natural step is a disadvantage.
> >
> > We also see simplicity as an advantage not a disadvantage! We explained why the safety state matters for the average cost-constrained problems and made it look natural. We think scheduling a safety budget is a simple idea and that is its strength. We presented two algorithms with this simple idea and again point to the other contributions. Further, this simple idea wasn't reported before hence perhaps it is worth publishing it for the benefit of the community. Perhaps, a more sophisticated approach to scheduling the safety budget can be derived, but even this algorithm works quite well.
> >
> > > I agree with Reviewer n7QX that the improvements resulting from empirical fine tuning on the given set of environments may be hard to generalize to a wider set of benchmarks. Hence it is important to give a theoretical case on why the method may outperform other approaches.
> >
> > We are slightly confused by this response since we've already answered to the reviewer n7QX that a theoretical justification is given (and was given) Section 2.2. We welcome further comments by the reviewer regarding this issue and which parts of the approach require further theoretical justification.
> >
> > > The current revised version of the paper differs significantly from the originally submitted version and should undergo a new review process. E.g., a new set of benchmarks appeared in the main text after further 'fine-tunning' was done by the authors.
> >
> > Again we can agree to disagree. We think the changes are appropriate for the rebuttal process that is meant to allow for changes to the paper.  Further, we are confused as **we thought that the reviewer requested to add these benchmarks to the paper**. We can also use previous figures and leave the new ones in Appendix and this will not change our message.
> >
> > > The current revision is in excess of 9 pages, which violates the revision rules (the 9-page limit still holds for the revision). Hence I have not analyzed all of the new content carefully.
> >
> > Apologies for the misunderstanding, we read the rules as we can use 10 pages for the rebuttal, we saw no indication that the 9-page limit holds. Otherwise, the authors would be submitting one more page that the reviewers didn't have a chance to read.
> >
> > In any case, we trimmed the paper down to 9 pages by removing a figure with environments (requested by another reviewer - they didn't see it as a major addition, but we will raise it in their response too) and a few figures with learning curves on safety gym benchamrks.
> >
> > > the clarity of the presentation should be improved and made more formal, e.g., the weird tables with the algorithm steps.
> >
> > We are very confused about the "the weird tables with the algorithm steps" statement. **The reviewer requested a formal algorithm environment in the paper -  we used a latex environment algorithm2e, which is fairly standard in our opinion.**  Recall that we had a plain text description with math formulas before.
> >
> > We also note that rejected papers can be submitted to AAAI with these reviews. Hence your effort in clearing up some confusion now could be helpful for future reviewers.

---

### Meta-Review · Area_Chair_vg6k · 2022-08-29

**Recommendation:** Accept
**Confidence:** Less certain

**Metareview:**

The authors develop a novel method for safe RL extending the work of Sootla et al (https://proceedings.mlr.press/v162/sootla22a/sootla22a.pdf) to deal with both expected and probability one constraints, demonstrating the utility of safety state aggregation in both scenarios. The authors validate their approach empirically and conduct careful experiments on several benchmark domains, showing gains from their approach relative to prior work.

Reviewers pointed out several issues in presentation and novelty relative to prior work that the authors addressed adequately in the rebuttal phase. Hence, I recommend acceptance.

**Award:**

No

---

### Decision · Program_Chairs · 2022-09-14

Accept